# A conserved mechanism drives partition complex assembly on bacterial chromosomes and plasmids

Roxanne E Debaugny[1], Aurore Sanchez[1,†], Jérôme Rech[1], Delphine Labourdette[2], Jérôme Dorignac[3], Frédéric Geniet[3], John Palmeri[3], Andrea Parmeggiani[3,4], François Boudsocq[1], Véronique Anton Leberre[2], Jean-Charles Walter[3,*] & Jean-Yves Bouet[1,**] iD

## Abstract

Chromosome and plasmid segregation in bacteria are mostly driven by ParAB*S* systems. These DNA partitioning machineries rely on large nucleoprotein complexes assembled on centromere sites (*parS*). However, the mechanism of how a few *parS*-bound ParB proteins nucleate the formation of highly concentrated ParB clusters remains unclear despite several proposed physico-mathematical models. We discriminated between these different models by varying some key parameters *in vivo* using the F plasmid partition system. We found that "Nucleation & caging" is the only coherent model recapitulating *in vivo* data. We also showed that the stochastic self-assembly of partition complexes (i) is a robust mechanism, (ii) does not directly involve ParA ATPase, (iii) results in a dynamic structure of discrete size independent of ParB concentration, and (iv) is not perturbed by active transcription but is by protein complexes. We refined the "Nucleation & caging" model and successfully applied it to the chromosomally encoded Par system of *Vibrio cholerae*, indicating that this stochastic self-assembly mechanism is widely conserved from plasmids to chromosomes.

**Keywords** DNA segregation; *Escherichia coli*; F plasmid; ParAB*S*; plasmid partition

**Subject Categories** DNA Replication, Repair & Recombination; Microbiology, Virology & Host Pathogen Interaction; Quantitative Biology & Dynamical Systems

**Mol Syst Biol. (2018) 14: e8516**

## Introduction

The segregation of DNA is an essential process for the faithful inheritance of genetic material. Minimalistic active partition systems, termed Par, ensure this key cell cycle step in bacteria (Baxter & Funnell, 2014) and archaea (Schumacher *et al*, 2015). Three main types of bacterial partition systems have been identified and classified by their NTPase signatures. Of these, the type I, also called ParAB*S*, is the only one present on chromosomes and the most widespread on low-copy-number plasmids (Gerdes *et al*, 2000). Each replicon encodes its own ParAB*S* system and their proper intracellular positioning depends on the interactions of the three ParAB*S* components: ParA, a Walker A ATPase; ParB, a dimer DNA binding protein; and *parS*, a centromere-like DNA sequence that ParB binds specifically. The ParA-driven mechanism that ensures the proper location and the directed segregation of replicons relies on the positioning of ParB*S* partition complexes within the nucleoid volume (Le Gall *et al*, 2016) and on a reaction diffusion-based mechanism (Hwang *et al*, 2013; Lim *et al*, 2014; Hu *et al*, 2017; Walter *et al*, 2017).

The centromere-like *parS* sites are located close to the replication origin on chromosomes and plasmids, and are typically composed of 16-bp palindromic motifs (Mori *et al*, 1986; Lin & Grossman, 1998). ParB binds with high affinity to its cognate *parS* as dimers (Hanai *et al*, 1996; Bouet *et al*, 2000). This serves as a nucleation point for assembling high molecular weight ParB-*parS* partition complexes, as initially seen by the silencing of genes present in the vicinity of *parS* (Lynch & Wang, 1995; Lobocka & Yarmolinsky, 1996). ParB binds over 10 Kbp away from *parS* sites for all ParAB*S* systems studied to date (Rodionov *et al*, 1999; Murray *et al*, 2006; Sanchez *et al*, 2015; Donczew *et al*, 2016; Lagage *et al*, 2016). This phenomenon, termed spreading, refers to the binding of ParB to centromere-flanking DNA regions in a non-specific manner. The propagation of ParB on DNA adjacent to *parS* is blocked by nucleoprotein complexes such as replication initiator complexes in the case of the P1 and F plasmids (Rodionov *et al*, 1999; Sanchez *et al*, 2015), or repressor–operator complexes on the bacterial chromosome (Murray *et al*, 2006). These "roadblock" effects led to the initial proposal that ParB propagates uni-dimensionally on both sides of the *parS* sites, in a so-called "1D-spreading" model (see

1  Laboratoire de Microbiologie et Génétique Moléculaires, Centre de Biologie Intégrative (CBI), Centre National de la Recherche Scientifique (CNRS), Université de Toulouse, UPS, Toulouse, France
2  LISBP, CNRS, INRA, INSA, Université de Toulouse, Toulouse, France
3  Laboratoire Charles Coulomb, CNRS-Université Montpellier, Montpellier, France
4  Dynamique des Interactions Membranaires Normales et Pathologiques, CNRS-Université Montpellier, Montpellier, France
   *Corresponding author. Tel: +33 467 143 146; E-mail: jean-charles.walter@umontpellier.fr
   **Corresponding author. Tel: +33 561 335 906; E-mail: jean-yves.bouet@ibcg.biotoul.fr
   †Present address:Institut Curie, UMR 3664 CNRS-IC, Paris, France

Fig EV1A). However, this model was put into question as (i) the quantity of ParB dimers present in the cell was insufficient to continuously cover the observed spreading zone, and (ii) ParB binding to *parS* adjacent DNA resisted biochemical demonstration (reviewed in Funnell, 2016).

As an alternative to "1D-spreading", two other models for partition complex assembly have been proposed, namely "Spreading & bridging" (Broedersz *et al*, 2014) and "Nucleation & caging" (Sanchez *et al*, 2015). Both models (see Fig EV1A) rely on strong ParB clustering with over 90% of ParB confined around *parS* (Sanchez *et al*, 2015). The "Spreading & bridging" model proposes that nearest-neighbor interactions (1D-spreading) initiated at *parS* and non-*parS* DNA sites in combination with their subsequent interactions in space (3D-bridging), lead in one of the conditions tested (strong spreading and bridging) to the condensation of the ParB-bound DNA into a large 3D complex over a contiguous 1D DNA domain (Broedersz *et al*, 2014; Graham *et al*, 2014). The "Nucleation & caging" model rather proposes that the combination of dynamic but synergistic interactions, ParB-ParB and ParB-nsDNA (Sanchez *et al*, 2015; Fisher *et al*, 2017), clusters most of the ParB around *parS* nucleation sites where a few ParB dimers are stably bound (Fig 1A). The *in vivo* ParB binding pattern from high-resolution ChIP-sequencing data was described with an asymptotic decay as a characteristic power law with an exponent b = $-3/2$, corresponding to the decreasing probability of the DNA to interact with the ParB cluster as a function of the genomic distance from *parS* (Sanchez *et al*, 2015). This model therefore proposes that the DNA surrounding the *parS* site interacts stochastically with the sphere of high ParB concentration. Interestingly, these three different assembly mechanisms have been explicitly modeled (Broedersz *et al*, 2014; Sanchez *et al*, 2015), thus allowing their predictions to be experimentally tested.

To study the assembly mechanism of partition complexes, we used the archetypical type I partition system of the F plasmid from *Escherichia coli*. By varying several key parameters, we evaluated ParB binding patterns *in vivo* in relation to predictions of each model. We also investigated the chromosomal ParABS system of the main chromosome of *Vibrio cholerae*. In all tested conditions, our data indicate that ParB binding profiles robustly correlate only with the predictions of the "Nucleation & caging" model.

## Results

### ParB$_F$ distribution pattern around *parS*$_F$ is similar on chromosome and plasmid DNA

The F plasmid partition complex assembles on a centromere sequence, *parS*$_F$, composed of twelve 43-bp tandem repeats (Helsberg & Eichenlaub, 1986), which contain ten 16-bp inverted repeat motifs to which ParB$_F$ binds specifically *in vitro* (Pillet *et al*, 2011) and *in vivo* (Sanchez *et al*, 2015). Partition complex assembly has been investigated using small versions of the F plasmid, either ~10 or ~60 Kbp. To discriminate between the different partition complex assembly models, we used two larger DNA molecules: the native 100-Kbp F plasmid (F1-10B; Appendix Table S1) and the 4.6-Mbp *E. coli* chromosome with *parS*$_F$ inserted at the *xylE* locus, in strains either expressing (DLT1472) or not (DLT1215) ParB$_F$ from an IPTG-inducible promoter.

We first verified the formation of ParB$_F$ clusters on these two different DNA molecules using the ParB$_F$-mVenus fluorescent fusion protein. ParB$_F$-mVenus, fully functional in plasmid partitioning (Appendix Table S2), was expressed from the endogenous locus on the F plasmid (F1-10B-BmV) or from a low-copy-number plasmid under the control of an IPTG-inducible promoter (pJYB294). In both cases, we observed bright and compact foci in nearly all cells (Fig 1B and D), indicating that the assembly of highly concentrated ParB$_F$ clusters on *parS*$_F$ from large DNA molecules, plasmid or chromosome, occurs similar to the smaller F plasmid counterparts (Sanchez *et al*, 2015). The number of foci from *parS*$_F$ inserted on the chromosome is half of what is observed with the F plasmid, as expected from the twofold difference in copy number (Collins & Pritchard, 1973).

We then performed ChIP-sequencing using anti-ParB antibodies and compared the ParB$_F$ patterns from the 100-Kbp F1-10B plasmid and the *xylE*::*parS*$_F$ chromosome insertion (ChIP-seq data are summarized in Table EV1). For F1-10B, we observed a ParB binding pattern extending over 18 Kbp of *parS*$_F$-flanking DNA nearly identical to the one previously observed on the 60-Kbp F plasmid (Sanchez *et al*, 2015), with the asymmetrical distribution arising from RepE nucleoprotein complexes formed on the left side of *parS*$_F$ on *incC* and *ori2* iterons (Fig 1C). Besides the strong ParB binding enrichment in the vicinity of *parS*$_F$, no other difference in the pattern between the input and IP samples was observed on the F plasmid and on the *E. coli* chromosome. When *parS*$_F$ is present on the chromosome, the ParB$_F$ binding pattern displays a comparable enrichment of *xylE*::*parS*$_F$-flanking DNA over 15 Kbp (Fig 1E). The ParB$_F$ distribution extends ~9 and 6 Kbp on the right and left sides of *parS*$_F$, respectively. The asymmetry does not depend on *parS*$_F$ orientation as an identical ParB$_F$ binding pattern was observed with *parS*$_F$ inserted in the reversed orientation (*xylE*::*parS*$_F$-rev, Appendix Fig S1B and C). Similar patterns were also observed when ParB$_F$ or ParB$_F$-mVenus were expressed *in trans* from a plasmid (Appendix Figs S1D and S3D). To the left side of *parS*$_F$, ParB$_F$ binding ends near the *yjbE* locus that harbors two promoters (locus A; Fig 1E, inset and Appendix Fig S1A), and to the right, ParB$_F$ binding ends at the *yjbI* gene locus (locus E; Fig 1E and Appendix Fig S1A). A dip in the ParB$_F$ binding intensity is also observed ~1 Kbp downstream from *parS*$_F$ spanning ~300 bp, corresponding to a promoter region (locus C; Fig 1E and Appendix Fig S1A). Dips and peaks in this ParB$_F$ binding pattern differ in terms of position and intensity when compared to the one present on the F plasmid. Overall, these data clearly indicate that the global ParB$_F$ binding distribution around *parS*$_F$ depends neither on the size nor the DNA molecule, plasmid or chromosome, and that the ParB$_F$ binding probability is dependent on the local constraints of each given locus.

### The "Nucleation & caging" binding model describes the partition complex assembly from the nucleation point to large genomic distance

Based on a smaller version of the F plasmid, we previously proposed the "Nucleation & caging" model describing ParB stochastic binding at large distance (> 100 bp) from *parS* due to DNA looping back into the confined ParB cluster. The characteristic asymptotic decay is compatible with a power law with the exponent b = $-3/2$, a property that is also observed with 100-Kbp F plasmid

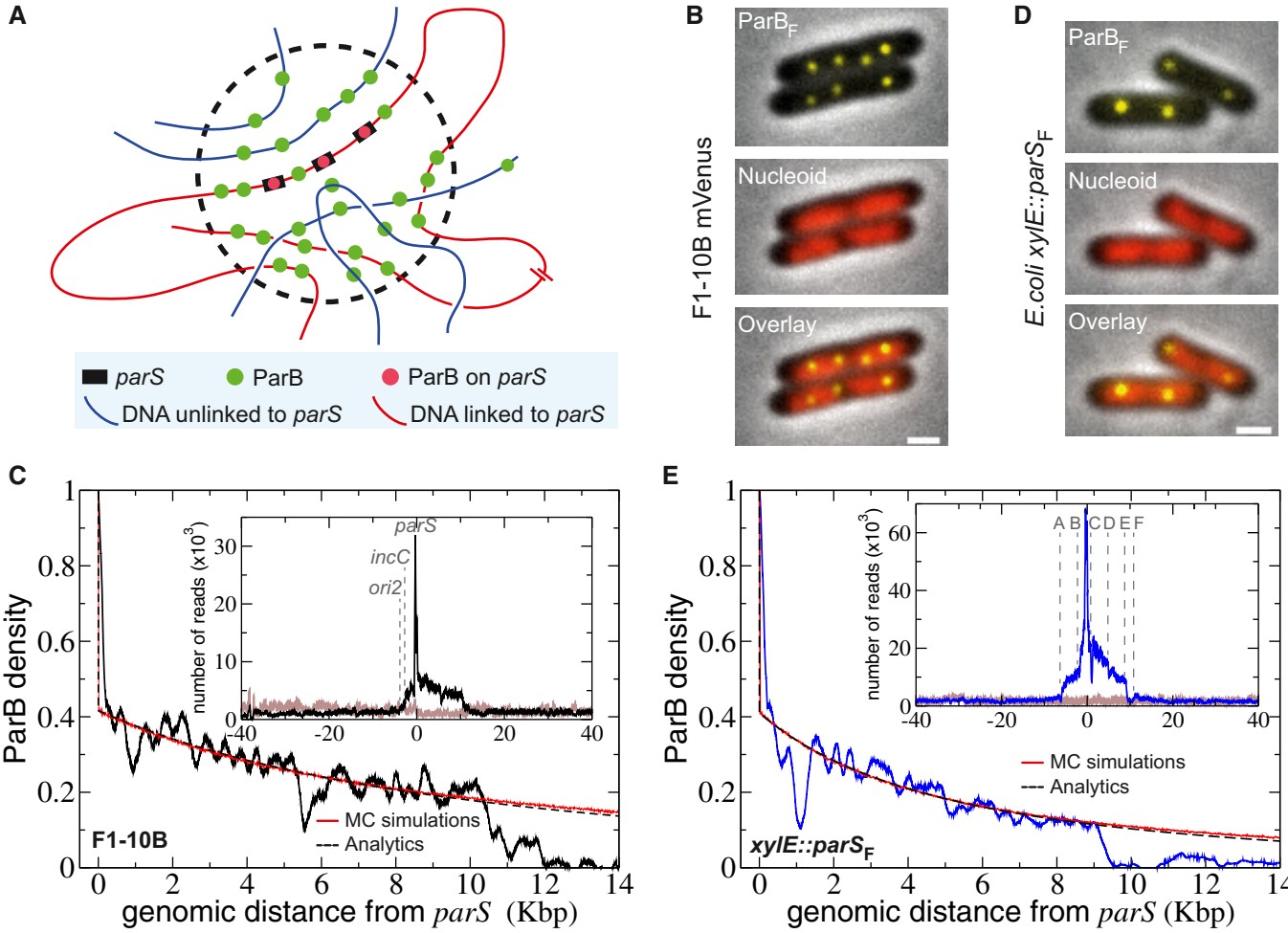

**Figure 1. ParB$_F$ binding outside of *parS* centromere on plasmid and chromosome.**

A    Schematic representation of the "Nucleation & caging" model. Most ParB dimers (green dots) are highly confined in a cluster (dotted circle) centered on the *parS* sites (black rectangles) onto which some ParBs are stably bound (red dots). The DNA entering the cluster is bound stochastically by ParB. Red and blue lines represent DNA present at small and large (or on a different molecule) genomic distance from *parS*, respectively.

B    ParB clusters on F plasmid *in vivo*. Typical *Escherichia coli* cells (DLT3594) display foci of ParB$_F$-mVenus protein (top) expressed from the endogenous genetic locus of the F plasmid (F1-10B-mVenus). The nucleoid is labeled with Hu-mCherry (central). The overlay (bottom) combines the two fluorescent channels. Over 99% of cells harbor ParB$_F$ foci. Scale bars: 1 μm.

C    ParB$_F$ binding outside *parS$_F$* on the F plasmid is compatible with a power law decay. High-resolution ChIP-seq performed on DLT3586 carrying the F plasmid (F1-10B). The ParB density, normalized to 1 at the first bp downstream the last *parS$_F$* binding repeat after background subtraction, is displayed over 14 Kbp on the right side of *parS$_F$*. Monte Carlo simulations and analytic formula are represented in red and dotted black lines, respectively. MC simulations were performed with a Freely Jointed Chain of linear length $L$ = 15 Kbp and a cluster radius $\sigma$ = 75 nm. The two other parameters, the Kuhn length $a$ = 10 bp and the total number of proteins on the F plasmid $N_t$ = 360 (related to the normalization constant of the protein concentration κ = 0.41), were fitted from the ChIP-seq data (see text and Box 1). As a benchmark for simulations, the analytics are obtained from equation (1) with the same parameters. *Inset*: The ParB$_F$ binding profile (black line) is represented as the number of nucleotide reads over 80 Kbp centered at *parS*. The number of reads in the input sample (gray line) is normalized to the total number of reads in the IP sample.

D, E    Same as (B and C) with *parS$_F$* inserted at the *xylE* locus on *E. coli* chromosome from DLT3584 and DLT2075, respectively. Cells were grown in the presence of 100 μM IPTG. The Kuhn length was adjusted to $a$ = 22 bp in the simulations and analytics. The characteristics of the A–F genetic loci are presented in Appendix Fig S1A. Note that a highly similar ParB$_F$ DNA binding pattern is obtained when ParB$_F$ was expressed *in trans* from a plasmid (strain DLT3567; Appendix Fig S1D).

(Fig 1C) and with *parS$_F$* inserted on the *E. coli* chromosome (Fig 1E and Appendix Fig S1C). This property is thus an intrinsic parameter of the ParB$_F$ binding profile at distance > 100 bp from *parS$_F$*. The abrupt initial drop in ParB$_F$ binding at a shorter genomic distance (< 100 bp) from *parS$_F$* is explained by the difference of ParB$_F$ binding affinities between specific *parS$_F$* sites ($K_d$ ~2 nM) and non-specific DNA ($K_d$ ~300 nM; Ah-Seng *et al*, 2009). We modeled the

DNA molecule by a Freely Jointed Chain (FJC) constituted of $N$ monomers of size $a$ [Kuhn length about twice the persistence length of the corresponding Worm-like chain (Schiessel, 2013)]. One particle is always attached on *parS* whereas non-specific sites are in contact with a reservoir of particles displaying a Gaussian distribution centered on *parS*. The ParB density was normalized to 1 by the value on the right side of *parS* and captured for non-specific sites in

the following phenomenological formula as the product of two probabilities integrated over the volume:

$$P_{NC}(s) = \int d^3 r P(r, s) C(r), \tag{1}$$

where $P(r, s) = \left(\frac{3}{2\pi R^2(s)}\right)^{3/2} e^{\frac{-3r^2}{2R(s)^2}}$ is the probability for two DNA loci spaced by a genomic distance $as$ to be at a distance $r$ in space for a Gaussian polymer (de Gennes, 1979); $R(s) = a\sqrt{s}$ is the equilibrium size of the section of DNA of linear length $as$; $C(r) = \kappa e^{\frac{-r^2}{2\sigma^2}}$ is the probability to find a protein ParB at a radial distance $r$ from the centromere, with $\kappa$ a normalization constant setting the total number $N_t$ of ParB on the DNA molecule and $\sigma$ the typical size of the cluster. Note that $C(r)$ is the linearized form of the Langmuir model (Phillips *et al*, 2012) offering a more compact and intuitive expression for $P_{NC}(s)$. From (1), we easily calculate (see Box 1 for the details of the calculation):

$$P_{NC}(s) = \frac{\kappa}{\left(\frac{a^2 s}{3\sigma^2} + 1\right)^{3/2}}. \tag{2}$$

Note that the decay versus the genomic distance $as$ is asymptotically determined by a power law of exponent $-3/2$ modulated by an amplitude depending on the concentration of ParB. The model has only three parameters: $\sigma = 75$ nm is determined from superresolution microscopy (Lim *et al*, 2014; Sanchez *et al*, 2015). The two remaining parameters $\kappa$ (a function of the total number of proteins $N_t$) and the Kuhn length $a$ are readily obtained from a fit of ChIP-seq data (see Box 1 for the calculation and Materials and Methods for the fitting procedure). Note that the relation between $\kappa$ and $N_t$ depends on the bioinformatics analysis (Appendix Fig S1E). We obtained $\kappa = 0.41$ for both F plasmid or $parS_F$-chromosomal insertions, leading to 360 and 120 ParB per DNA molecule, respectively, in good agreement with former estimate (Bouet *et al*, 2005). The last remaining free parameter is the Kuhn length $a$, estimated to 10 or 22 bp for the F plasmid or $parS_F$-chromosomal insertions, respectively, to fully describe the $ParB_F$ DNA binding profiles (Fig 1C and E, and Appendix Fig S1D). These fitted values are lower than expected, likely due to the modeling that does not account for supercoiling and confinement. Nevertheless, using these defined parameters, the refined "Nucleation & caging" model provides a qualitative prediction of the experimental data over the whole range of genomic positions, from a few bp to more than 10 Kbp.

## $ParB_F$ DNA binding pattern over a wide range of ParB concentrations favors the "Nucleation & caging" model

The physical modeling for each proposed model (Broedersz *et al*, 2014; Sanchez *et al*, 2015) predicts distinct and characteristic responses upon variation of the intracellular ParB concentration (see explanations in Fig EV1B). Briefly, (i) the "1-D filament" model predicts a rapid decrease of ParB binding followed by a constant binding profile dependent on ParB amount, (ii) the "Spreading & bridging" model predicts linear decays with slopes depending on the ParB amount, and (iii) the "Nucleation & caging" model predicts a binding profile which depends only on the size of the foci. The exponent b = $-3/2$ of the power law distribution would not change upon ParB amount variation resulting in an overall similar decay at

**Box 1: Analytic calculation of the linear probability of bound particles along DNA**

We model the DNA molecule by a Freely Jointed Chain (FJC) characterized by $N$ freely rotating monomers of size $a$ (total linear length $L = aN$). The probability distribution $P(r, s)$ to have two monomers of a Gaussian polymer at a distance $r$ and spaced by $s$ monomers (linear distance $as$) along the polymer is given by de Gennes (1979):

$$P(r, s) = \left(\frac{3}{2\pi R^2(s)}\right)^{3/2} e^{\frac{-3r^2}{2R(s)^2}}, \tag{3}$$

where $R(s) = a\sqrt{s}$ is the averaged radius occupied by a portion of polymer of size $as$. In the same way, we define the probability to find a particle ParB at the distance $r$ from $parS$ with a Gaussian repartition centered at $parS$ and with a width $\sigma$ corresponding to the averaged radius of the foci occupied by proteins:

$$C(r) = \kappa e^{\frac{-r^2}{2\sigma^2}}, \tag{4}$$

where $\kappa$ is an adimensional normalization constant setting the total number of ParB on the DNA. Thus, the occupation rate of a protein on DNA is given by:

$$P_{NC}(s) = \int d^3 r P(r, s) C(r) = \int dr 4\pi r^2 P(r, s) C(r). \tag{5}$$

The integration of equation (5) gives:

$$P_{NC}(s) = \frac{\kappa}{\left(\frac{a^2 s}{3\sigma^2} + 1\right)^{3/2}}. \tag{6}$$

Note that $P_{NC}(0) = \kappa$, thus $\kappa$ is setting the height of the drop between specific and non-specific sites and can be estimated directly from the ChIP-seq data. When $R^2(s) \gg 3\sigma^2$, we recover a pure algebraic law $P_{NC} \sim s^{-3/2}$. The total number of particle $N_t$ on the plasmid is:

$$\int_0^N ds P_{NC}(s) = N_t. \tag{7}$$

The latter integral gives the expression of the parameter $\kappa$ as a function of $N_t$:

$$\kappa = \frac{1}{2} \frac{1}{3^{3/2}} \left(\frac{a}{\sigma}\right)^3 \frac{N_t}{1/(\sqrt{3}\sigma/a) - 1/\sqrt{N + 3(\sigma/a)^2}} \xrightarrow[N\to\infty]{} \frac{1}{6}\left(\frac{a}{\sigma}\right)^2 N_t. \tag{8}$$

The second term in equation (8) containing the total number $N$ of monomers induces only corrections to the dominant behavior, we will thus restrict ourselves to the length enriched in ChIP-seq, i.e. 15 Kbp. Note that the limit $N\to\infty$ in equation (8) gives us a condition on the ratio $a/\sigma$ in order to have proteins on DNA. As $\kappa$ is the amplitude of a probability, it has to satisfy the condition $0 \le \kappa \le 1$. Indeed, at a fixed $\sigma$, if the Kuhn length becomes too large the polymer does not return in the focus frequently enough in order to ensure $N_t$ bound proteins onto the DNA.

We note that the ParB proteins that bind to the DNA molecule targeted by ChIP-seq come from a bound state on competing non-spe DNA (see Fig 1A). Thus, the gain in energy is zero and the binding is solely governed by entropy. However, regarding the binding on specific DNA, there is a gain of energy corresponding to the difference between specific and non-specific binding energies $\Delta\varepsilon = \varepsilon_s - \varepsilon_{ns}$, respectively. This energy difference $\Delta\varepsilon$ is sufficiently large in *E. coli* to consider that $parS$ sites are always occupied.

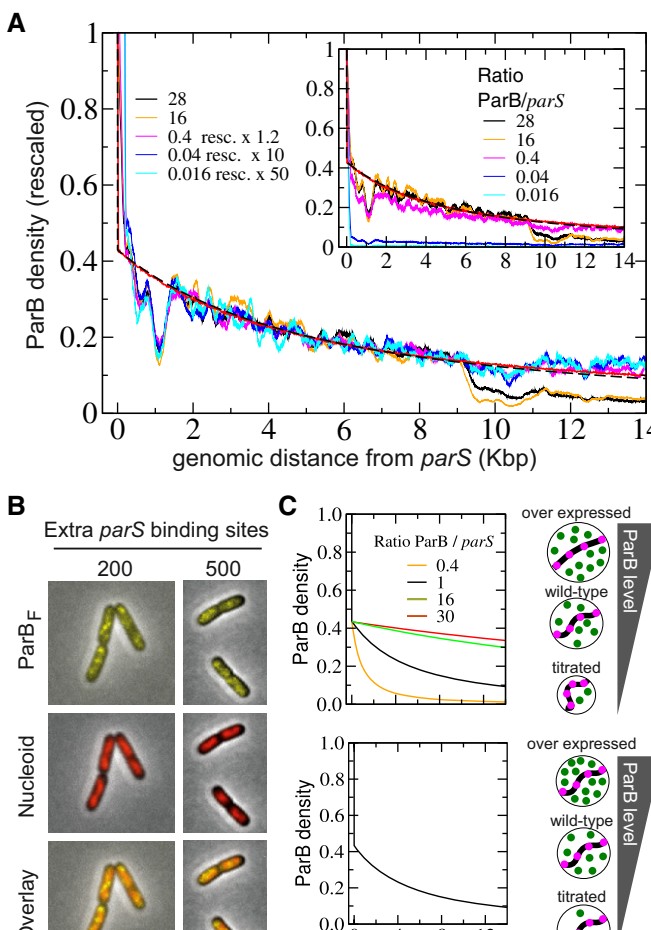

**Figure 2. ParB_F DNA binding pattern is robust over a large range of intracellular ParB_F concentrations.**

A   Normalized and rescaled ParB_F binding profiles at different ParB_F/parS_F ratio. ChIP-seq density on the right side of parS_F inserted at xylE was measured in DLT2075 induced (16, 28) or not (0.4) with IPTG (100 and 500 μM), or carrying HCN plasmids pZC302 (0.04) or pJYB57 (0.016), normalized as in Fig 1C and E with the amplitudes of the curves rescaled by the indicated factors (1.2, 10, or 50) to overlap with the curves of highest amplitude. The ParB/parS ratio is calculated relative to the one of F plasmid as determined from Western blot analyses (Appendix Fig S2B). Monte Carlo simulations and analytical formula are plotted with the same parameters as in Fig 1E. Note (i) that the dips at ~9 Kbp are not visible for the low levels of available ParB since the signal is close to the basal level, and (ii) that the ChIP-seq data at 100 μM IPTG induction (16) are the same as in Fig 1E. *Inset*: Same as in the main to display the density without rescaling.

B   ParB_F is dispersed in the cell upon titration by HCN plasmids. ParB_F-mVenus expressed from pJYB294 was imaged as in Fig 1D in DLT3577 (left) and DLT3576 (right) carrying pZC302 and pJYB57, respectively. The number of extra parS_F per cell, indicated on top of each raw, is estimated from the copy number per cell of HCN plasmids carrying 10 specific binding sites. Scale bars: 1 μm.

C   The size of ParB_F clusters is independent of the intracellular ParB_F concentration. We considered two possible evolutions of the cluster size upon variations of ParB amount in the framework of "Nucleation & caging" with corresponding schematics drawn on the right. For direct comparison with (A), all curves are displayed with a rescaling of the amplitude corresponding to the WT expression level. *Top*: constant ParB concentration; supposing that clusters are compact, the cluster radius $\sigma$ would depend on the number $m$ of ParB like $\sigma = m^{1/3}$. Predictions profiles, plotted at different ratio of ParB/parS, vary within the range of the experimental levels tested. *Bottom*: constant cluster size; ParB concentrations vary but the range of exploration remains the same resulting in overlapping profiles.

a fixed focus size. In order to discriminate between these three model predictions, we performed ChIP-seq experiments over a large range of intracellular ParB concentrations. To prevent interference with plasmid stability, we used the chromosomally encoded xylE::parS_F construct expressing parB_F under the control of an IPTG-inducible promoter (DLT2075).

Without IPTG induction, ParB_F was expressed at ~0.2 of the physiological concentration from F plasmid, as judged by Western blot analyses (Appendix Fig S2B). We also tested an 8- and 14-fold overproduction of ParB_F. Assuming the twofold difference in copy number (Fig 1B and D), these three conditions provided ParB_F/parS_F ratios of 0.4, 16, and 28, relative to the F plasmid one. At these three ratios, ChIP-seq data revealed that ParB_F binding extended similarly over ~15 Kbp around parS_F. We analyzed the right side of parS_F displaying the longest propagation distance by normalizing each dataset (Fig 2A). It revealed that regardless of ParB_F concentration, (i) the ParB_F distribution in the vicinity of parS_F always displays a good correlation with a power law fitting with an exponent of $-3/2$, (ii) the ParB_F binding profile ends at the same genomic location, i.e. 9 Kbp from parS_F, and (iii) the location of the dips and peaks in the pattern is highly conserved, as confirmed by correlation analyses (Table EV2). These findings indicate a highly robust ParB_F binding pattern that is invariant over a ~70-fold variation of the ParB_F amount.

To further vary the amount of ParB_F available for partition complex assembly, high-copy-number (HCN) plasmids containing the parS_F sequence were introduced into the xylE::parS_F strain to efficiently titrate ParB_F by its binding to the excess of specific binding sites (~200- and ~500-fold on pBR322 and pBSKS derivatives, respectively; Diaz *et al*, 2015). Epifluorescence microscopy of these strains reveals that all cells display a diffuse ParB-mVenus fluorescence (Fig 2B) in contrast to concise foci without titration (Fig 1A), suggesting a large reduction of ParB availability to non-specific sites in the vicinity of parS_F on the chromosome. ChIP-seq analyses in the two titration conditions revealed that ParB binding in the vicinity of parS_F was dramatically reduced as expected. However, rescaling the signals by a factor of 10 and 50 for the pBR322 and pBSKS parS_F-carrying derivatives, corresponding to a ParB_F/parS_F ratio of 0.04 and 0.016, respectively, revealed a ParB_F binding pattern above the background level (Fig 2B, inset). In both datasets, ParB_F binding decreases progressively over about the same genomic distance and with a similar power law decay as without titration. Moreover, even with these very low amounts of available ParB_F, the dips and peaks in the profiles are present at similar positions (Table EV2).

The invariance of the overall ParB profile over three orders of magnitude of ParB concentration (Fig 2B, inset) excludes the predictions of both the "1-D filament" and the "Spreading & bridging" models (Fig EV1). In addition, the conservation in the positions of the dips and peaks indicates that the probability of ParB_F binding at a given location is also not dependent on the amount of ParB_F in the clusters. These results are strongly in favor of the refined "Nucleation & caging" model presented above.

## The size of the dynamic ParB/parS cluster is independent of ParB intracellular concentration

In all of the ParB induction levels tested, the genomic distance over which ParB$_F$ binds around parS$_F$ is constant and displays a very similar decay (Fig 2A). This conserved binding behavior could provide information on the cluster size as a function of ParB amount. Indeed, the "Nucleation & caging" model predicts a density per site $P_{NC}(s) = \kappa \left( \frac{a^2 s}{3\sigma^2} + 1 \right)^{-3/2}$ (see equation 2). Thus, the $P_{NC}(s)$ decay is entirely determined by the geometry of the foci and the intrinsic flexibility of the DNA, and the overall amplitude depends on the number of ParB. Varying the ParB amount could lead to two limiting situations: (i) the density of ParB, but not $\sigma$, is constant, (ii) $\sigma$ is fixed and ParB density is variable. We plotted these two situations in the range of ParB/parS ratio considered experimentally (Fig 2C): with (i) the different $P_{NC}(s)$ strongly varied, and (ii) $P_{NC}(s)$ was invariant relative to the ParB amount resulting in overlapping profiles. Experimental data (Fig 2A) are in excellent agreement with the latter. From this modeling, we thus concluded that the size of partition complexes is invariant to change in ParB intracellular concentration.

## The arginine rich motif (box II) of ParB$_F$ is critical for partition complex assembly

The ability of ParB to multimerize through dimer–dimer interactions is required for the formation of ParB clusters. A highly conserved patch of arginine residues present in the N-terminal domain of ParB (box II motif; Yamaichi & Niki, 2000) has been proposed to be involved in ParB multimerization (Breier & Grossman, 2007; Song et al, 2017). To examine to what extent the box II motif is involved in vivo in the assembly of ParB$_F$ clusters, we changed three arginine residues to alanine (Appendix Fig S3A). The resulting ParB$_F$-3R* variant was purified and assayed for DNA binding activity by electro-mobility shift assay (EMSA) in the presence of competitor DNA using a DNA probe containing a single parS$_F$ site (Fig 3A). ParB$_F$-3R* binds parS$_F$ with high affinity (B1 complex) indicating no defect in parS binding nor dimerization, a property required for parS binding (Hanai et al, 1996). However, in contrast to WT ParB, the formation of secondary complexes (B'2 and B'3), resulting from non-specific DNA binding and dimer–dimer interaction (Sanchez et al, 2015), was impaired further suggesting the implication of box II in dimer–dimer interaction. A mini-F carrying the parB$_F$-3R* allele (pAS30) was lost at a rate corresponding to random distribution at cell division (Appendix Table S2), indicating that this variant is unable to properly segregate the mini-F.

The ParB$_F$-3R* variant was then expressed in native or fluorescently tagged (ParB-R3*-mVenus) forms, from pJYB303 or pJYB296, respectively, in the xylE::parS$_F$ strain. By imaging ParB$_F$-3R*-mVenus, we observed only faint foci in a high background of diffuse fluorescence (Fig 3B). These barely detectable foci may correspond to ParB$_F$-3R*-mVenus binding to the 10 specific sites present on parS$_F$ and, if any, to residual ParB$_F$ cluster formation. We then performed ChIP-seq assays with ParB$_F$-3R* present in ~25-fold excess (relative ParB$_F$/parS$_F$ ratio compared to the F plasmid one; Appendix Fig S3B). The resulting DNA binding profile displayed

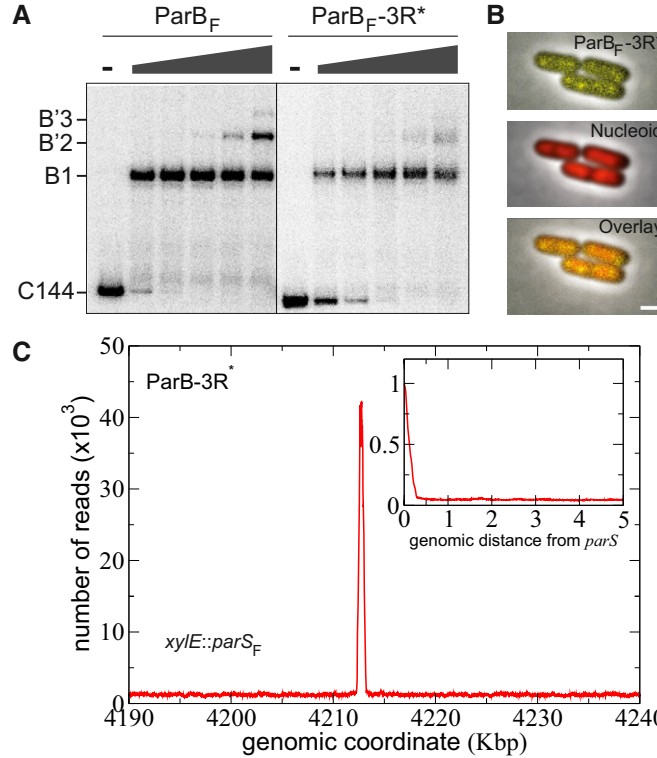

**Figure 3. The box II motif of ParB$_F$ is crucial for ParB$_F$ binding in the vicinity of parS$_F$ and cluster formation.**

A  The formation of secondary ParB$_F$-DNA complexes requires the box II motif. EMSA was performed with a 144-bp [32]P-labeled DNA fragments (C144) carrying a single 16-bp parS binding motif. Reaction mixtures containing 100 µg ml$^{-1}$ sonicated salmon sperm DNA were incubated in the absence (−) or the presence of increasing concentrations (gray triangle; 10, 30, 100, 300, and 1,000 nM) of ParB$_F$ or ParB$_F$-3R*. Positions of free and bound probes are indicated on the left. B1 represents complexes involving the specific interaction on the 16-bp binding site, while B'2 and B'3 complexes represent secondary complexes involving the parS$_F$ site with one or two additional nsDNA-binding interactions, respectively (Sanchez et al, 2015).

B  ParB$_F$ cluster formation requires the box II motif. Epifluorescence microscopy of ParB$_F$-3R*-mVenus from DLT3566 is displayed as in Fig 1D. Scale bars: 1 µm.

C  ParB$_F$ in vivo DNA binding in the vicinity of parS$_F$ sites requires the box II motif. ChIP-seq was performed on DLT3726 carrying parS$_F$ in the xylE chromosomal locus and expressing ParB$_F$-3R* variant. ParB$_F$-3R* DNA binding profile displayed the number of nucleotide reads as a function of the Escherichia coli genomic coordinates. The peak at parS$_F$ covered approximately 950 bp, which corresponds to the 402 bp between the 1[st] and 10[th] specific binding sites and ~280 bp on each sides (representing the average size of the DNA library; see Appendix Fig S3E). No ParB$_F$-3R* enrichment was found on parS$_F$-flanking DNA and elsewhere on the chromosome. Inset: Zoom in on the right side of parS$_F$ over 5 Kbp with the ParB density, normalized to 1 at the first bp after the last parS binding repeat, plotted as a function of the distance from parS$_F$. Note that a highly similar DNA binding pattern is obtained with ParB$_F$-3R*-mVenus (strain DLT3566; Appendix Fig S3C).

enrichment only at parS$_F$ with a total absence of ParB$_F$ binding on parS$_F$-flanking DNA (Fig 3C). Indeed, no residual ParB$_F$ binding to non-specific DNA was detected when the size of the DNA fragments in the IP library is taken into account (Appendix Fig S3E). This pattern differs from those observed in conditions of ParB$_F$ titration

(Fig 2A; inset), indicating that the ParB$_F$-3R* box II variant is fully deficient in clustering *in vivo*. The same patterns were also observed with ParB$_F$-3R*-mVenus (Appendix Fig S3C and F) indicating that the mVenus fluorescent-tag fused to ParB$_F$ does not promote cluster assembly.

Together, these results indicate that the box II variant is specifically deficient in ParB$_F$ cluster assembly but not in *parS*$_F$ binding, and thus reveal that the box II motif is critical for the auto-assembly of the partition complex.

### ParB also propagates stochastically from native chromosomal *parS* sites

ParAB*S* systems are present on most bacterial chromosomes (Gerdes *et al*, 2000). To determine whether chromosomal ParB-*parS* partition complexes also assembled *in vivo* in a similar manner to the F plasmid, we investigated the bacterium *V. cholerae*, whose genome is composed of two chromosomes. We focused on the largest chromosome to which ParB$_{Vc1}$ binds to three separated 16-bp *parS* sites comprised within 7 Kbp (Saint-Dic *et al*, 2006; Baek *et al*, 2014; Fig 4A).

We purified ParB$_{Vc1}$ antibodies against his-tagged ParB$_{Vc1}$ and performed ChIP-seq assays on exponentially growing cultures. The ParB$_{Vc1}$ DNA binding pattern covered ~18 Kbp and displayed three peaks at the exact location of the three *parS*$_{Vc1}$ sites (Fig 4B). No other ParB binding was observed over the *Vibrio* genome. Each peak exhibits a distinct but reproducible difference in intensity that might correspond to the slight differences in *parS*$_{Vc1}$ sequences (Appendix Fig S4A). An asymmetry in the binding pattern was observed on the left side of *parS*1 with the limit of ParB$_{Vc1}$ binding corresponding to the end of the rRNA operon located ~4 Kbp upstream from *parS*1 (Fig 4B). This suggests that highly transcribed genes might significantly interfere with the extent of ParB binding.

We modeled ParB$_{Vc1}$ DNA binding profile with the framework of the refined "Nucleation and caging" model (see above). The simulations consider three non-interacting spheres centered on each of the *parS* sites and take into account (Fig 4C) the average fragment size of the DNA library to account for the width of the peaks around each *parS* (same modeling as displayed in Appendix Fig S1E for the F plasmid). Simulations are found in good agreement with the ChIP-seq data with the following parameters: $\sigma$ = 25 nm, $a$ = 16 bp, and $\kappa$ = 0.15 leading to $N_t$~50 ParB proteins on the chromosome (see Materials and Methods for the fitting procedure). Overall, these parameters are of the same order of magnitude as those used for *E. coli*. The maxima in the ParB binding profile depends on the *parS* sites (Fig 4C) and are interpreted as a difference in binding affinity. In the simulations, the ParB density is normalized to 1 by the value on the right of *parS*1. The relative density of the two other p*arS* sites is fixed according to the values read on the ChIP-seq plot (3 and 29% lower affinity for *parS*2 and *parS*3 compared to *parS*1, respectively). We also noticed a clear difference at the minima of ParB binding on either side of *parS*2 (64.2 and 68 Kbp; Appendix Fig S4B). In the case of a single cluster constraining the three *parS*, the profile would only depend on the genomic distance from *parS*2 resulting in a symmetrical pattern, while in the case of three independent

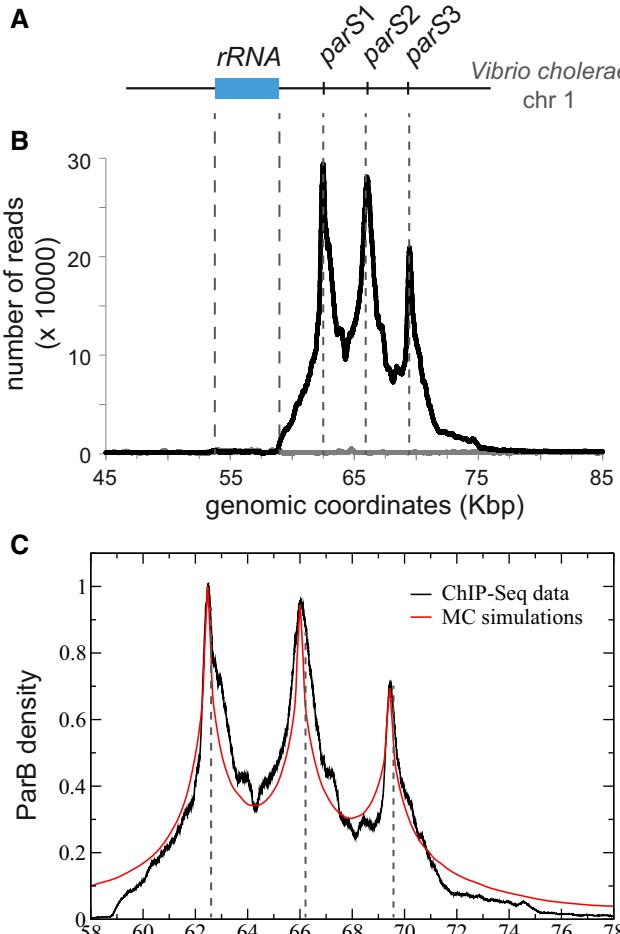

**Figure 4.  ParB of *Vibrio cholerae* assembled in cluster similarly to ParB$_F$.**

A    Schematic representation of the genomic locus of the chromosome 1 of *V. cholerae* with the three *parS* sites, named *parS*1-3. The rRNA operon (blue rectangle) spans the genomic coordinates 53,823–59,123.

B    ChIP-seq performed on strain N16961 is displayed as the number of nucleotide reads in function of the genomic coordinates. Correspondence to the *parS*1-3 location represented in (A) is indicated by gray dotted lines. The number of reads in the input sample (gray line) is normalized to the total number of reads in the IP sample.

C    We modeled the ChIP-Seq data as in Fig 1C–E by means of MC simulations with a Freely jointed chain of size $N$ = 2,000 monomers of size $a$ = 16 bp. Data are normalized after background subtraction to the read value at *parS*1 (genomic coordinate 62,438). The best fit was achieved with $\sigma$ = 25 nm and an amplitude $\kappa$ = 0.15 leading to $N_t$~50 ParB on the chromosome. In the MC simulation, we accounted for the finite width of the distribution around *parS* sites by including the average fragment size of the DNA library (304 bp; for comparison, a simulation without is provided in Appendix Fig S4B).

clusters, an absence of symmetry due to the occupation of the specific sites is expected. This indicates that the system displays three independent clusters nucleated at each *parS* sites. However, the possibility that these clusters mix together at a frequency dependent on the genomic distance between *parS* sites is not excluded. At larger distances from *parS* sites, differences between the experimental data and the simulation probably arise

from strong impediments to ParB binding, such as the presence of the rRNA operon.

These data strongly support that the partition complex assembly mechanism is conserved on plasmid and chromosome ParAB*S* systems.

## Nucleoprotein complexes, but not active transcription, are the major determinants for the impediment of ParB stochastic binding

The major dips in the ParB$_F$ DNA binding signal are often found at promoter loci (Appendix Fig S1A). To investigate the link between gene expression and the impediment to ParB propagation, we reproduced the ChIP-seq assays using the *xylE*::*parS*$_F$ strain grown in the presence of rifampicin, an inhibitor of RNA synthesis that traps RNA polymerases at promoters loci in an abortive complex unable to extend RNAs beyond a few nucleotides (Herring *et al*, 2005). We did not observe significant changes to the ParB signal on either side of *parS*$_F$ (Fig 5A; compare red and blue curves). Notably, the ParB signal still strongly drops in promoter regions (e.g., loci A, C, and E) and the dips and peaks are present at the same locations (Fig 5B and Table EV2). This indicates that active transcription by RNA polymerase is not a major impediment to ParB binding, but rather that RNA polymerases bound or stalled at the promoter could.

We also measured the ParB binding profile in stationary phase, a growth condition in which gene expression is strongly reduced. On the right side of *parS*$_F$, ParB distribution was similar to all other tested conditions (Fig 5A), thus confirming the robustness of the binding pattern. On both sides, the strong reduction of ParB binding at loci A, C, and E was still observed. However, in contrast to the other conditions, ParB binding recovers after these loci and extends up to ~18 Kbp on both sides, resulting in the location of *parS*$_F$ in the

middle of a ~36 Kbp propagation zone. Interestingly, the ParB binding profiles after these recoveries are still compatible with a power law exhibiting the same characteristics as at lower genomic distances (Fig 5C). In stationary phase, the reduced intracellular dynamics

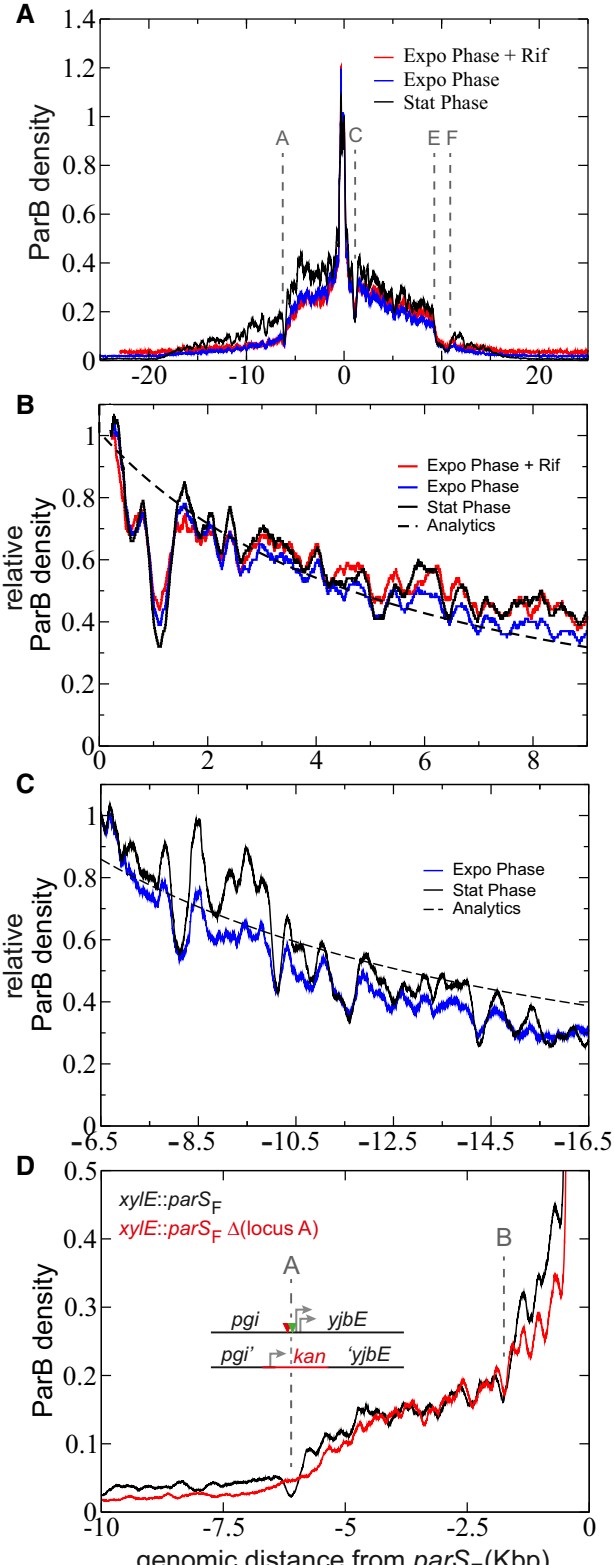

**Figure 5. Robust dips and peaks signatures in ParB DNA binding profiles.**

ChIP-sequencing assays were performed on DLT2075 (*xylE*::*parS*$_F$) expressing ParB$_F$ grown in exponential (expo) or stationary (stat) phases with addition of rifampicin when indicated (+Rif). The assays have been performed in duplicate for the +Rif and once for the stationary phase experiments.

A    ParB$_F$ DNA binding around *parS*$_F$ is independent of active transcription. The color-coded ParB$_F$ profiles are represented over 50 Kbp as the relative ParB density normalized to 1 at the first bp after the last *parS*$_F$ binding site. Loci A, C, E, and F are defined in Appendix Fig S1A.

B    The dips and peaks are highly similar in the three indicated conditions. Same as in (A) with zoom in on the right side of *parS*$_F$ up to 9 Kbp and normalization to 1 at genomic coordinate 230. The dotted line corresponds to the analytics description of "Nucleation and caging" (see details in Fig 1C–E).

C    ParB$_F$ binding profile upstream of the locus A. Same as in (A) with zoom in from −6.5 to −16.5 Kbp by normalization to 1 at genomic coordinate −6.5 Kbp (upstream of the dip at the locus A). The ParB$_F$ DNA binding profile remains compatible to a power law, represented by the analytics description (dotted line), upstream of the locus A in stationary phase (black) and in exponential phase (blue). Also, the dips and peaks are highly similar in both conditions. These data are not in favor of the "1D-spreading" or the "Spreading and bridging" models that predicts a basal uniform distribution or a linear decrease after a barrier, respectively (Broedersz *et al*, 2014).

D    The promoter region at locus A prevents ParB$_F$ DNA binding. ChipIP-seq assays were performed in isogenic *xylE*::*parS*$_F$ strains (DLT2075; black curve) in which the locus A is replaced by a kanamycin gene (DLT3651; red curve). The assay in the Δ(locus A) genomic context has been performed once. The relative ParB density as a function of the distance from *parS*$_F$ is drawn and normalized as in (A). The promoter region is depicted as in Appendix Fig S1A.

(Parry *et al*, 2014) and the higher compaction of the DNA (Meyer & Grainger, 2013) may stabilize the partition complex revealing the ParB$_F$ bound at larger distances from *parS*$_F$. Also, in higher (stationary phase) or lower (rifampicin-treated cells) DNA compaction states (Appendix Fig S5A), the ParB$_F$ DNA binding pattern is not altered, exhibiting a similar profile of dips and peaks (Fig 5B). This indicates that the assembly of the partition complex is not perturbed by variation in DNA compaction level within the nucleoid.

To further demonstrate the impediment of ParB$_F$ binding in promoter regions, we constructed a strain in which the locus A, carrying two promoters, an IHF and two RcsB binding sites, is replaced by a kanamycin resistance gene (Fig 5D). The measured ParB$_F$ binding pattern remained highly comparable except at the locus A where the dip is absent. This result clearly indicates that site-specific DNA binding proteins are the main factors for restricting locally ParB$_F$ binding.

### ParB molecules exchange rapidly between partition complexes

Single molecule *in vivo* localization experiments have shown that over 90% of ParB$_F$ molecules are present at any time in the confined clusters (Sanchez *et al*, 2015). However, stochastic binding of most ParB$_F$ on non-specific DNA suggests that partition complexes are highly dynamic. To unravel ParB$_F$ dynamics, we performed fluorescence recovery after photobleaching (FRAP) on two-foci cells for measuring ParB$_F$ dynamics between partition complexes. By laser-bleaching only one focus, we could determine whether ParB$_F$ dimers could exchange between clusters and measure the exchange kinetics. As ParB$_F$ foci are mobile, we choose to partially bleach (~50%) the focus enabling immediate measurement of fluorescence recovery (Fig 6A and B). A few seconds after bleaching, the fluorescence intensity recovers while it decreases in the unbleached focus. This exchange is progressive and the intensity between the two foci equilibrated in ~80 s on average (between 50 and 120 s for most individual experiments). We estimate that, when exiting a cluster, each ParB$_F$ dimer has the same probability to reach any of the two clusters. Therefore, the time of equilibration between the two foci corresponds to the exchange of all ParB$_F$. These results thus indicate that the partition complexes are dynamic structures with a rapid exchange of ParB$_F$ molecules between clusters.

## Discussion

Despite over three decades of biochemical and molecular studies on several ParAB*S* systems, the mechanism of how a few ParB bound to *parS* sites can attract hundreds of ParB in the vicinity of *parS* to assemble a high molecular weight complex remained puzzling. The three main mechanisms proposed for ParB-*parS* cluster assembly have been studied from physico-mathematical perspectives (Broedersz *et al*, 2014; Sanchez *et al*, 2015), predicting very different outcomes for the ParB binding profile in the vicinity of *parS* sites upon change in ParB concentration. Here, the ParB binding patterns were found invariant over a large variation of ParB amount displaying a robust decay function compatible with a power law with the characteristic exponent b = −3/2 and a conserved length of the propagation zone (Fig 2A). Strikingly, even in the titration conditions tested, which resulted in a very low amount of ParB available to bind to

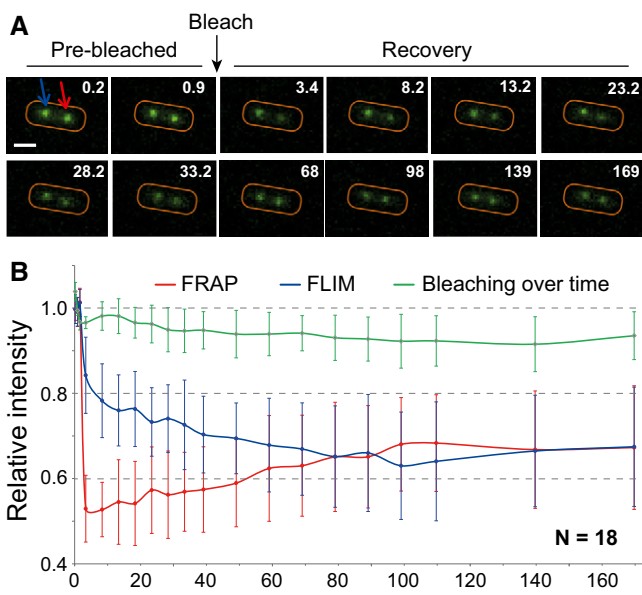

**Figure 6. ParB dynamics between partition complexes.**

ParB$_F$ exchange between foci was measured by FRAP and FLIM (fluorescence lifetime imaging microscopy) from two-foci cell of DLT1215 carrying pJYB234.

A   Representative images of a photobleached cell during a FRAP experiment. The 488 nm laser was pulsed (Bleach) on one of the two foci at ~2.4 s (black arrow). Red and blue arrows correspond to the bleached and unbleached focus, respectively. Time is indicated in seconds (upper right). The cell outline is drawn in red. Scale bar: 1 μm.

B   Quantification of ParB$_F$-mVenus fluorescence intensity over time. The dynamics of fluorescence intensity is shown from averaging 18 independent measurements of the bleached (FRAP, red line) and unbleached (FLIM, blue line) foci. Foci fluorescence intensity in each experiment was normalized to the average intensity of each focus before photobleaching. The Appendix Fig S6 displays the three pre-bleaching and the first post-bleaching data points on an expanded scale. Natural bleaching during the course of the experiments (green curve) was estimated for each measurement by averaging the fluorescence intensity of 15 foci present in each field of view. Error bars correspond to standard deviation (mean ± SD).

non-specific DNA sites, the overall ParB DNA binding pattern remained invariant (Fig 2A, inset). Neither "1D spreading" nor "Spreading & bridging" physical models could describe these data in the conditions tested (Broedersz *et al*, 2014). A variant of the latter model has explored the ParB binding pattern in the low spreading strength limit (Walter *et al*, 2018). This "Looping & clustering" model also predicts variations in the ParB binding pattern over a simulated 4-fold range of ParB amount, which is in contrast to the invariant pattern observed experimentally over more than three orders of magnitude (Fig 2). In conclusion, only the "Nucleation & caging" model based on stochastic ParB binding well describes the experimental data and provides accurate predictions for the mechanism of the partition complexes assembly.

We refined the modeling of the dynamic and stochastic ParB binding model by including DNA binding affinities for specific and non-specific sites to describe the initial drop observed immediately after *parS*. In this framework, we found that ParB clusters have a constant size accommodating important variations in ParB concentration (Fig 2C). We propose that the cluster size is dependent on the intrinsic ParB-ParB and ParB-nsDNA interactions, and would

thus be an inherent characteristic of each ParAB*S* system (Funnell & Gagnier, 1993; Sanchez *et al*, 2015; Taylor *et al*, 2015). In the case of F and P1 plasmids, overexpression of ParB was reported to silence genes in the vicinity of their cognate *parS* (Lynch & Wang, 1995; Lobocka & Yarmolinsky, 1996), by a mechanism based on 1D-spreading (Rodionov *et al*, 1999). Our finding that the size of ParB clusters is invariant but their density increases with ParB overexpression provides a new explanation for the silencing phenomenon. We propose that RNA polymerases accessibility to promoters present near *parS* is dependent on the ParB density within the cluster. At physiological level, RNA polymerase would have efficient access to promoter sites while upon the rise of ParB level their diffusion within the high-density cluster would be reduced proportionally to the overexpression level, as observed experimentally (Rodionov *et al*, 1999). This is reminiscent to the observation that a change in the level of supercoiling is specifically induced on ParB-*parS* carrying mini-F plasmids (Biek & Shi, 1994). It has been shown that this deficit in negative supercoiling could be due to the reduce accessibility of DNA gyrases to the small sized mini-F plasmid (< 10 Kbp) that is "masked" by the ParB-*parS* nucleoprotein complex (Bouet & Lane, 2009). The invariance in the size of the ParB cluster but the density may also well explain the supercoiling deficit observed *in vivo*. The refined modeling also well describes the chromosomal partition system of *V. cholera*, predicting three independent clusters nucleated at each of the three *parS* sites (Fig 4C). In all cases reported here, the partition complex assembly is well described by the "Nucleation & caging" model, and we propose that this mechanism of assembly is conserved on chromosome and plasmid partitioning systems.

In addition to its robustness within a large range of ParB concentration (Fig 2A) and different nucleoid compaction states (Fig 5A), the *in vivo* ParB DNA binding pattern also exhibits conserved dips and peaks at particular locations (Table EV2). The major dips are located at promoter regions (Fig 1E and Appendix Fig S1A) but do not depend on active transcription (Fig 5B). This suggests that these specific signatures mostly depend on the intrinsic local genomic environment. This hypothesis was confirmed by deleting the locus A, carrying several regulator binding sites, which led to the suppression of the dip at this position (Fig 5D). Therefore, proteins such as transcriptional regulators and NAPs (nucleoid-associated proteins) that bind specifically to DNA prevent ParB binding to these sites, thus reducing locally the ParB signal. We propose that this impediment to ParB binding is proportional to the time of occupancy of these regulators at their site-specific DNA binding sites. Larger nucleoprotein complexes, as exemplified on the F plasmid at the iteron sites (*ori2* and *incC*; Fig 1C) that interact *in cis* and *in trans* (Das & Chattoraj, 2004), were previously proposed to be spatially excluded from the vicinity of the ParB cluster with a low probability that DNA beyond these sites comes back into the cluster preventing ParB binding (Sanchez *et al*, 2015). Such an exclusion does not occur from smaller protein–DNA complexes, with the recovery of the ParB binding signal that further follows the characteristic power law decay (e.g., locus A; Fig 5C). These results show that low molecular weight protein–DNA complexes do not impair the overall, only the local, ParB binding pattern.

The formation of highly concentrated clusters of ParB relies on a strong ParB-*parS* interaction and two other interactions, ParB-ParB and ParB-nsDNA (Sanchez *et al*, 2015; Fisher *et al*, 2017). ParB

mutants that do not propagate outside *parS* are impaired in partition activity and in cluster formation *in vivo* (Rodionov *et al*, 1999; Breier & Grossman, 2007). The conserved box II motif (Yamaichi & Niki, 2000) was suggested to be part of the dimer–dimer interface (Breier & Grossman, 2007; Graham *et al*, 2014) but some misfolding caveat has been reported with some mutants, such as ParB$_{Bsub}$-G77S (Song *et al*, 2017). *In vivo* the box II variant (ParB$_F$-3R*) is totally deficient in partition activity and cluster formation (Fig 3B) while proficient for *parS*$_F$ binding (Fig 3C). The total absence of ParB$_F$-3R* binding outside *parS*$_F$ (Fig 3C and Appendix Fig S3E) indicates that the box II motif is the major interface for the interaction between ParB dimers and is critical for the partition complexes assembly *in vivo* and the DNA partition activity.

ParA interacts with partition complexes in a ParB-dependent manner both *in vitro* and *in vivo* (Bouet & Funnell, 1999; Lemonnier *et al*, 2000) to ensure the ATP-dependent segregation of centromere sites upon DNA replication (Fung *et al*, 2001; Scholefield *et al*, 2011; Ah-Seng *et al*, 2013). Previous studies from *V. cholerae* and *S. Venezuela* have reported contradictory results on the involvement of ParA in the assembly of the partition complex (Baek *et al*, 2014; Donczew *et al*, 2016), which may arise from the pleiotropic effects of ParA on cellular processes, such as gene transcription or DNA replication (Murray & Errington, 2008). The ParB$_F$ DNA binding profiles on the F plasmid (Fig 1C) and on the *E. coli* chromosome (Fig 1E), in the presence and absence of ParA$_F$, respectively, are highly similar, therefore indicating that they assemble independently of ParA. Partition complexes, composed of hundreds of ParB dimers, were thought to be confined at the interface between the nucleoid and the inner membrane (Vecchiarelli *et al*, 2012). The observation that they rather are located within the nucleoid in a ParA-dependent manner (Le Gall *et al*, 2016) raises the question as to how they are not excluded from it. The "Nucleation & caging" model could solve this apparent paradox. Indeed, relying on a strong ParB-*parS* interaction (nM range) and two other synergistic, but labile interactions, ParB-ParB and ParB-nsDNA (hundreds of nM range; Fisher *et al*, 2017; Sanchez *et al*, 2015), it would allow the dynamic confinement of most ParB without forming a rigid static structure. This dynamic organization is further supported by the finding that ParB dimers quickly exchange between clusters (~80 s; Fig 6). By comparison, the equilibration times between H-NS and TetR-*tetO* clusters were 5 or 10 times much longer, respectively (Kumar *et al*, 2010). Since > 90% of ParB are present in clusters (Sanchez *et al*, 2015), it implies that their time of residency is much longer inside than outside, in agreement with fast diffusion coefficients (~1 μm$^2$ s$^{-1}$) for non-specific DNA binding proteins (Kumar *et al*, 2010). We propose that, collectively, all the individual but labile interactions for partition complex assembly allow the whole complex attracted by ParA to progress within the mesh of the nucleoid.

## Materials and Methods

### Bacterial strains and plasmids

*Escherichia coli* and *V. cholerae* strains and plasmids are listed in Appendix Table S1. Cultures were essentially grown at 37°C with aeration in LB (Miller, 1972) containing thymine (10 μg ml$^{-1}$) and

chloramphenicol (10 μg ml$^{-1}$) as appropriate. For microscopy and stability assays, cultures were grown at 30°C with aeration in MGC (M9 minimal medium supplemented with 0.4% glucose, 0.2% casamino acids, 1 mM MgSO$_4$, 0.1 mM CaCl$_2$, 1 μg ml$^{-1}$ thiamine, 20 μg ml$^{-1}$ leucine, and 40 μg ml$^{-1}$ thymine).

Strain DLT1471 was constructed in several steps. First, the *Eco*47III–*Apa*I *sopOPAB* DNA fragment from plasmid F was inserted into a *rep*$^{ts}$ plasmid-borne 933-codon 3′ fragment of the *lacZ* gene (pJYB50). The resulting plasmid, pJYB52, was introduced into DLT1215 and proceeded for allele exchange by selecting double recombination events as described (integration–excision assay; Cornet *et al*, 1994), yielding to the following chromosome fusion *lacZ'*::*PparAB*$_F$::*'lacZ* in strain DLT1471. Strain DLT1472 expressing ParB$_F$ from the chromosome fusion *lacZ'*::*PparB*$_F$::*'lacZ* was described previously (Bouet *et al*, 2005).

Strains DLT2073 and DLT2075 are DLT1215 and DLT1472 derivatives, respectively, in which a 538-bp DNA fragment carrying the entire *parS*$_F$ site has been introduced at the *Eco*RV restriction site of the *xylE* gene (91 min on the *E. coli* chromosome) in the forward orientation. Briefly, the PCR-amplified *xylE* gene was introduced onto a pFC13 derivative (Cornet *et al*, 1994) using the *Hind*III and *Xho*I restrictions sites, leading to pJYB102. A *parS*$_F$ DNA fragment, PCR-amplified from pDAG114 using the following oligonucleotides SopC-5′RV (5′-TCCTTTGATATCGGCCAGAAAGCATAACTG-3′) and SopC-3′RV (5′-GCCGATATCAGGAATTCATGGAATCGTAGTCTC-3′), was introduced into the *Eco*RV restriction of the *xylE* locus on pJYB102, leading to pJYB103.1 and pJYB103.2 depending on the orientation of *parS*$_F$ insertion. These plasmids were introduced into DLT1472 and subjected to the integration–excision procedure as above. Strains DLT2074 and DLT2076 are identical to DLT2073 and DLT2075, respectively, with *parS*$_F$ inserted in *xylE* in the reverse orientation (*parS*$_F$-rev). Insertions of *parS*$_F$ on the *E. coli* chromosome at locations other than *xylE* were performed using lambda red recombineering and selecting for the FRT-Kan$^R$-FRT cassette amplified from pDK4 (Datsenko & Wanner, 2000). The *hupA-mcherry* allele used for live imaging of the nucleoid was inserted in various *E. coli* strains by P1 transduction from strain DLT3053 (Le Gall *et al*, 2016).

The original plasmid F, F1-10 (gift from C. Lesterlin), was converted to *ccdB*$^-$ to allow for performing stability assays. The removal of the CcdB toxin from the addiction system was performed along with the introduction of the chloramphenicol resistant gene, using lambda red recombineering (Datsenko & Wanner, 2000), by inserting the corresponding loci from the mini-F pDAG114 (Lemonnier *et al*, 2000) into F1-10, leading to F1-10B. When necessary, the excision of the FRT-kan-FRT selection cassette was performed using the pCP20 plasmid (Datsenko & Wanner, 2000). F1-10B ΔAB and F1-10B-BmV derivatives were constructed by lambda red recombineering using plasmids pDAG209 and pJYB234, respectively, as substrates to generate the linear DNA fragment that include the *cat* gene for selecting the recombinants.

Mutations in *parB*$_F$ were first introduced into pYAS6 by mutagenic primer-directed replication using the Stratagene QuikChange kit and subsequently integrated into mini-F derivatives or the pAM238 expression vector by PCR amplification followed by allelic replacement using appropriate restriction enzymes. The *mVenus* gene was constructed by introducing the monomeric A207K mutation in the *venus-Yfp* gene (Sanchez *et al*, 2015). All plasmid

constructs were verified by DNA sequencing (MWG). Mini-F and F plasmids derivatives were introduced in strains by CaCl$_2$- or electro-transformation and conjugation, respectively.

pJYB322 was constructed by introducing a 171-bp DNA fragment carrying three consensus *parS* sites (5′-TGTTTCACGTGAAACA-3′), called 3x-*parS*$_{chr}$, into the *Nhe*I and *Cla*I restriction sites of pJYB263, the Δ*parA* derivative of pJYB234 (Le Gall *et al*, 2016).

## Plasmid stability assays

Stability of mini-F and plasmid F derivatives was assayed in strain DLT1215 grown over 25 generations in MGC at 30°C or over 20 generations in LB at 37°C, and subsequently plated on LB agar medium, replica plating to medium with chloramphenicol, and calculating loss rates from the fractions of each sample resistant to chloramphenicol, as previously described (Sanchez *et al*, 2013).

## Epifluorescence microscopy

Exponentially growing cultures were deposited on slides coated with a 1% agarose buffered solution and imaged as previously described (Diaz *et al*, 2015), using an Eclipse TI-E/B wide field epifluorescence microscope. Snapshots were taken using a phase contrast objective (CFI Plan Fluor DLL 100X oil NA1.3) and Semrock filters sets for YFP (Ex: 500BP24; DM: 520; Em: 542BP27) and Cy3 (Ex: 531BP40; DM: 562; Em: 593BP40) with an exposure time range of 0.1–0.5 s. Nis-Elements AR software (Nikon) was used for image capture and editing.

## ChIP-sequencing assay and analysis

ChIP-seq was performed as previously described (Diaz *et al*, 2017) with minor modifications, using polyclonal antibodies raised against WT ParB$_F$ or his-tagged ParB$_{Vc1}$. ParB$_F$ and ParB$_{Vc}$-1 antibodies were affinity-purified from anti-ParB$_F$ (our own) and anti-ParB$_{Vc}$-1 (gift donated from the lab of D. Chattoraj) polyclonal serums, using purified ParB$_F$ and ParB$_{Vc}$-1-his$_6$, respectively. The optimization step for determining the amount of antibodies needed to pull down all ParB$_{Vc1}$ in the IP samples was fully described (Diaz *et al*, 2017).

A maximum of 2 ng of ChIP DNA was used to synthetize the library as described in manufacturer's instructions (Ion ChIP-Seq Library Preparation on the Ion Proton™ System—revision B—Step10). The last size selection was modified by a double size selection (0.55×/0.25×) of binding to AMPure® XP beads followed by a step of wash and elution. After qualification and quantification (Bioanalyzer—Agilent, Santa Clara, CA), libraries were diluted to 100 pM and pooled in a ratio of 20% input and 80% IP. For all the subsequent analyses, the measured size distribution of the DNA fragments was subtracted of the linker size added for sequencing (93 bp). Template preparation (clonal library amplification on sequencing bead) was made using the Ion PI™ Template OT2 200 Kit version 3 following the manufacturer's instructions. Emulsion PCR was performed in the Ion OneTouch™ 2 Instrument. DNA-positive ISPs were then recovered and enriched in the Ion OneTouch™ ES according to standard protocols. Sequencing of samples was conducted on Ion Proton and PI chips according to the Ion PI™ 200 Sequencing Kit Protocol (version 3).

The sequence reads were counted and aligned as described (Diaz *et al*, 2017). The quality of reads was assessed with FastQC program (https://www.bioinformatics.babraham.ac.uk/projects/fastqc/) and they were mapped with TMAP from Torrent Suite software 5.0.4 (https://www.thermofisher.com/fr/fr/home/life-science/sequencing/next-generation-sequencing/ion-torrent-next-generation-sequencing-workflow/ion-torrent-suite-software.html). The counting of reads over the genomic sequences was performed using bedtools 2.26.0 (http://bedtools.readthedocs.io/en/latest/) with the tool genomecov and the option –d and –fs or –bga for the bedgraph files production for peak visualization. This allows to display the totality of ChIP-seq data with the possibility to view several datasets simultaneously using IGV (Integrated Genome Viewer, version 2.3; http://software.broadinstitute.org/software/igv/). Graphing the DNA portion of interest from ChIP-seq data was done using Excel or R softwares. Cognate input and IP samples were normalized by the number of total reads for direct comparison. For the ParB density plots, the data were normalized after background subtraction and set to the value of 1 at the last bp of the $10^{th}$ repeat of $parS_F$, allowing to display the results of Monte Carlo simulations on the same graph.

## Fit of the parameters

The "Nucleation and caging" model contains only three parameters: $\sigma$, $a$, and $N_t$. One parameter is known experimentally: $\sigma \sim 75$ nm (Lim *et al*, 2014; Sanchez *et al*, 2015). We know the order of magnitude of $N_t \sim 300$ (Bouet *et al*, 2005) as a benchmark of the fitted value obtained from $\kappa$ from ChIP-seq data. The value of $\sigma^2/a$ is fitted from ChIP-seq, which gives access to the value of $a$ ($\sigma$ being known). We use the following trial function to perform a non-linear fit of the ChIP-seq data between 0 and 10 Kbp:

$$P_{NC}(s) = \frac{A_0}{(A_1 a s + 1)^{3/2}},$$

where $A_0$ and $A_1$ are the two fit parameters allowing to obtained the two free parameters of the model: $\kappa$ (related to $N_t$) and $a$. Note that the length unit to analyze the ChIP-seq data is the base pair (bp). We identify $A_0 = \kappa$ and $A_1 = a/3\sigma^2$. For the F plasmid, we find $A_0 = 0.41$ and $A_1 = 1.96 \times 10^{-4}$ nm$^{-1}$. From $A_1$, using $\sigma = 75$ nm, we get $a \sim 3.4$ nm (~10 bp). From $A_0$, we get $\kappa \sim 0.41$ leading to $N_t \sim 360$. For the chromosome, using the same analysis, we get $A_0 = \kappa \sim 0.41$ leading to a smaller number of proteins on the DNA, $N_t \sim 120$, yet of the same order of magnitude as the F plasmid. The second variable of the fit $A_1 = 4.33 \times 10^{-4}$ nm$^{-1}$ leads to $a \sim 7.5$ nm (~22 bp). For the estimation of the parameters for *V. cholerae*, the experimental values for $\sigma$ and $a$ are not available, and $\kappa$ (or $N_t$) could not be read directly from the ChIP-seq data because of the fragment size library and the presence of barrier (rRNA operon) that impede the ParB binding signal at large genomic distance. Thus, given the order of magnitude observed in *E. coli* (between 10 and 20 bp for the F plasmid and the chromosome, respectively), it is reasonable to assume a same range for *V. cholerae*, with $a = 16$ bp corresponding to the footprint of a ParB. In order to conserve the height and peaks of the ChIP-seq data, we take $\sigma = 25 \pm 5$ nm and $\kappa = 0.15 \pm 0.05$. The probability to form a foci is chosen to be $P_{parS1} = 1$, $P_{parS2} = 0.9$, and $P_{parS3} = 0.6$ in order to match the height of each peak observed in ChIP-seq. These

simulations are only semi-quantitative, in order to show on very general physical ground that the "Nucleation and caging" is able to explain the long range decay observed in ChIP-seq.

## Monte Carlo simulations

The Monte Carlo procedures used an explicit polymer modeled by a Freely Jointed Chain (FJC). The polymer is in contact with a ParB reservoir. To reproduce the confinement of ParB around *parS*, the particles are not simulated explicitly. Instead, we modeled ParB binding to the polymer with a probability decreasing as a Gaussian function of the locus distance from *parS*. The Monte Carlo procedures were performed using the following scheme:

1. Build a Freely Jointed Chain (FJC) of $N$ monomers of size $a$.
2. Define a particular site on the FJC as *parS* (or potentially many *parS* for *V. cholerae*) and define a Gaussian distribution of particles $C(r) = \kappa e^{\frac{-r^2}{2a^2}}$.
3. For each monomer label by an index $i$, choose a random number *ran*. If $ran < C(r)$, then a particle is attached to the site $i$.
4. Start again at step 1 until the statistics is good enough.

This fast procedure allows to get very good statistics. In Figs 1C–E and 2A, and Appendix Fig S4B, we have analytical expressions for $P_{NC}(s)$, which serve as a benchmark for simulations at the bp resolution. In Fig 4C ($xylE::parS_F$), Appendix Fig S1E (plasmid F), and Appendix Fig S4B (*V. cholerae*) where we included the average DNA fragments size in the ChIP libraries—as well as three *parS* sequences for *V. cholerae*—the analytical expressions were not obvious, we therefore used simulations.

## Modeling the ChIP-seq data with the integration of the average fragments size of the DNA library

The ChIP-seq assay is based on the sequencing of sonicated fragments (of size ~250 bp) with an intrinsic unknown on the precise location of ParB (footprint of 16 bp). Here, the bioinformatic convention to build the ParB profile is to count +1 read at each bp of the fragments with at least one detected bound ParB. From the simulation perspective, this leads to two averages: (i) over the bound ParB positions, and (ii) over all fragment positions. When a bound ParB is detected, the average over the fragments positions consists in adding to the simulated profile a triangular function centered at the actual ParB position (where it takes the value of 1) and decreasing linearly down to 0 at a genomic distance $\pm$ the fragment size.

## Correlation analysis

The correlation analyses were performed using the following formula:

$$Correl(x, y) = \frac{\sum (x - \bar{x})(y - \bar{y})}{\sqrt{\sum (x - \bar{x})^2 \sum (y - \bar{y})^2}}.$$

## Western immunoblotting

The determination of ParB$_F$ relative intracellular concentrations and antibody purifications was performed as described (Diaz *et al*,

2015). When indicated, samples were diluted in DLT1215 extract to keep constant the total amount of proteins.

### EMSA and proteins purification

Electro-mobility shift assay was performed as described (Bouet *et al*, 2007) in the presence of sonicated salmon sperm DNA as competitor (100 mg ml$^{-1}$), using 1 nM radiolabeled 144-bp DNA probe containing a single *parS*$_F$ site generated by PCR. ParB$_F$ and ParB$_F$-3R* proteins were purified using an intein strategy as previously described from plasmids pYAS6 and pYAS25, respectively (Ah-Seng *et al*, 2009). ParB$_{Vc}$-1-his$_6$ was affinity-purified in a single step using a 10–1,000 nM imidazole gradient from strain DLT3431 (Castaing *et al*, 2008).

ParB$_F$ and ParB$_F$-3R* proteins were purified as previously described (Ah-Seng *et al*, 2009).

### FRAP and FLIM assays

*Escherichia coli* cells (Stellar) carrying pJYB213 (ParBF-eGfp), grown in mid-exponential phase in MGC medium and spotted on microscope slides coated with 1% MGC-buffered agarose, were subjected to laser-bleaching. Each field was imaged three times (pre-bleached step) before photobleaching (at ~2.4 s) a single ParB$_F$ focus with a 488 nm laser into two-foci cells. ROI (region of interest) were 0.2 × 0.2 or 0.3 × 0.3 corresponding to 5 or 9 pixels, respectively. The laser power was set between 67 and 74 Hz to ensure partial bleaching, thus enabling to follow fluorescence recovery on a time scale of second right after photobleaching. Images were taken using an EMCCD camera with a 0.13 μm per pixel resolution (Hamamatsu). To follow recovery dynamics, images were taken every 5, 10, and 20 s up to 38, 108, and 169 s, respectively. Overall, photobleaching in the field of view during the time course of each FRAP experiment (green curves in Fig 6B and Appendix Fig S6A) was averaged from 15 unbleached foci from each field. Normalization to 1 was performed by averaging the focus fluorescence intensity from the three pre-bleached images.

## Data availability

The datasets produced in this study are available in the following databases:

- Chip-Seq data for *V. cholera*: Gene Expression Omnibus GSE114980 (https://www.ncbi.nlm.nih.gov/geo/query/acc.cgi?acc = GSE114980)
- Chip-Seq data for *E. coli*: Gene Expression Omnibus GSE115274 (https://www.ncbi.nlm.nih.gov/geo/query/acc.cgi?acc = GSE115274)

Expanded View for this article is available online.

## Acknowledgements

We thank Y. Ah-Seng for the constructing the ParB$_F$-3R* allele, the platform GeT-Biopuces (Genopole, Toulouse) for sequencing experiments and S. Cantaloube and T. Mangeat (LITC-CBI platform) and C. Reyes for advices and preliminary experiments in microscopy. We are grateful to F. Cornet, P. Rousseau, P. Polard, M. Nollmann, I. Junier, and members of the team for fruitful discussions and critical reading of the manuscript. We thank C. Lesterlin for sharing the F1-10 plasmid, D. Chattoraj for the anti-ParB$_{Vc1}$ serum, and Y. Yamaichi for pSM836 and *V. cholerae* strains. This work was supported by Agence National pour la Recherche (ANR-14-CE09-0025-01) and the CNRS INPHYNITI program, RD by a PhD grant from Université de Toulouse (APR14), AP, FG, JP, and JCW by the Labex NUMEV (AAP 2013-2-005, 2015-2-055, 2016-1-024).

## Author contributions

Conceptualization, J-YB and AP; Methodology, J-YB, J-CW, VAL, and AP; Investigation, RED, J-CW, AS, JR, and J-YB; Formal Analysis, J-CW, JD, FG, JP, and AP; Writing – Original Draft, RED, J-CW, and J-YB; Writing – Review & Editing, J-YB, J-CW, RED, and AP; Funding Acquisition, J-YB, AP, JP, and VAL; Resources, DL and FB; Supervision, J-YB, VAL, and AP.

## Conflict of interest

The authors declare that they have no conflict of interest.

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
