## [Review Process File · Molecular Systems Biology]

A conserved mechanism drives partition complex assembly on bacterial chromosomes and plasmids

Roxanne E. Debaugny, Aurore Sanchez, Jérôme Rech, Delphine Labourdette, Jérôme Dorignac, Frédéric Geniet, John Palmeri, Andrea Parmeggiani, François Boudsocq, Véronique Anton Leberre, Jean-Charles Walter and Jean-Yves Bouet.

Review timeline:

Submission date:	22 nd June 2018
Editorial Decision:	20 th July 2018
Revision received:	13 th September 2018
Editorial Decision:	24 th September 2018
Revision received:	18 th October 2018
Accepted:	22 nd October 2018

Editor: Maria Polychronidou

Transaction Report:

1st Editorial Decision

20th July 2018

Thank you again for submitting your work to Molecular Systems Biology. We have now heard back from the three referees who agreed to evaluate your study. As you will see below, the reviewers think that the study could be a useful contribution to the field. They raise however a series of concerns, which we would ask you to address in a major revision.

In sum, the reviewers think that further analyses and additional controls are required to convincingly support the conclusions of the study. All three reviewers make constructive suggestions in this regard. I think the recommendations of the reviewers are clear and there is therefore no need to repeat the comments listed below. Please feel free to contact me in case you would like to discuss in further detail any of the issues raised by the reviewers.

REFeree REPORTS

Reviewer #1:

In this work, the authors provide further evidence to support their Nucleation and Caging model of partition complex formation and argue that the mechanism is widely conserved. The main results are:

-The ParB density profile (as a function of genomic distance from parS) for different intercellular ParB concentrations collapse onto one another. This is not captured by previous models but is by their model if the size of the partition complexes (ParB foci) is independent of the intercellular ParB concentration.

- Residues in the N-terminal domain are required for partition complex formation but not parS binding.
 - Nucleoprotein complexes, but not active transcription, impede ParB binding. ParB complexes are dynamic and turn over on a 1 min timescale.
 - Consistent ParB profiles are observed when parS is inserted into the chromosome and in around the native chromosomal parS sites of *V. cholera*.
- These results are generally well supported by the data and would make a useful contribution to the field. However, I have some comments that I believe should be addressed before publication.

Major points:

- In the authors' model, ParB binds and unbinds DNA and itself stochastically. There is no mention of 1D spreading. Yet they show that nucleoprotein complexes act as a barrier to ParB spreading (eg. *incC*, locus A and B in Fig. 1). If ParB is dynamic, why does its density not recover after the obstacle and thereby produce an approximately symmetric binding pattern? Indeed, in stationary phase, in which transcription is greatly reduced, the authors nicely show that the signal is much more symmetric. Connected with this, the statement that 'Nucleoprotein complexes, but not active transcription, are the major determinants for the impediment of ParB stochastic binding' is, to me, misleading. Does the data (exp vs exp+rif vs stationary, Fig. 5) not indicate that RNA polymerase acts as roadblock, whether involved in active transcription or stalled by rifampicin? Might gene orientation also play a role? The authors could check how gene orientation correlates the position of peaks and dips. It would be interesting to know whether this would indicate some role for 1-D spreading.
- The authors' model makes an interesting and testable prediction i.e. that the size of ParB foci is independent of the intracellular concentration. This perhaps could be tested experimentally using standard fluorescence microscopy (in the model the authors use a value coming from super-resolution microscopy) and would provide stronger support for their model. I realise that additive effects may mean that a brighter focus appears wider than it actually is but such effects may not be significant over a moderate range of intensities and it may be worth a try.
- The description of the Monte Carlo simulations is severely lacking. All I could find was in the legend to Fig. 1. This should be explained in full and it should be made clear how ParB interacts with the polymer and the effect thereon. Related to that, the model includes ParB-ParB interactions but it wasn't clear to me if ParB is treated explicitly in the simulations or simply enforced externally as a fixed spatial gradient.

Minor points:

- How much IPTG is used in Fig 1E?
- As the inset in Fig. 2A is arguable more important than the outer figure, the authors may want to swap the two around. Perhaps correlation analysis would help demonstrate the locations of peaks and dip coincide.
- Is the orange line in Fig. 2A the same as in Fig. 1E?
- It is stated in the text that the densities plotted in Fig. 2A are only of the right side of profile. This should be stated in the legend and it seems to also be the case for Fig 1C,E and Fig S1 but is not stated.
- Why do the 3 curves with lowest ParB ratio not have the dip at 9kbp in Fig. 2A?
- p14, p16 The authors should be careful in their wording regarding Power law fitting. It is one thing to say the analytic (power law) relation is consistent with the data and it is another to say that the data exhibits a power law behaviour. It is very easy to obtain a good fit of experimental data to a power law and caution is required. See Stumpf & Porter, Science 2012, DOI: 10.1126/science.1216142
- Reference for the form of $P(r,s)$ in equation 1.
- 'F plasmid' instead of 'plasmid F'
- p10. 'model predicts a probability ..' -> 'model predicts a density ..'
- p13 What is the Stochastic Binding model? Is it the same as the Nucleation and Caging model? One name would suffice.
- 'stochastic binding'. All binding is stochastic, whether slow or fast. This is not a useful phrase. I think the authors rather mean that binding/unbinding occurs on relatively fast timescale i.e. foci are not static but are turned over continuously.
- p15. That 90% of ParB are in clusters, does not at all imply that partition complexes are stable structures. They could still be turned over slowly or rapidly.

Reviewer #2:

This study examines the *in vivo* mode of assembly of plasmid and chromosomal partition complexes in bacteria. It has been known for a long time that many ParB protein molecules bind at and around the *parS* partition site, form large complexes that are visible as foci by different fluorescence imaging approaches, and spread many kb outward into non-specific DNA away from the *parS* sites by ChIP and ChIP-seq approaches. The authors have previously proposed the "nucleation and caging" model, and here they further address this model in relation to a second "spreading and bridging" model (the 1D spreading models originally proposed have been previously ruled out by these and other authors). The authors have refined the caging model, and a new contribution in this work is that they test it by examining ParB patterns as a function of intracellular ParB concentration (F plasmid ParB), both experimentally and mathematically. They find that the patterns of ParB binding/spreading are independent of ParB concentration, which fits their modeling of caging "robustly" and best. Another new contribution is an examination of the "dips" in ParB binding on the spreading ChIP-seq curves. It is known that strong protein binding sites provide "roadblocks" to ParB spreading, but the curves outside of the roadblocks are not smooth. Some dips correlate with promoters, and the authors show that (i) they do not need active transcription, and (ii) are absent when the promoter region is removed, for example.

The study also confirms ParB spreading properties that have been examined in other systems, which, while not new, are important contributions to show that these properties are general ones for these types of ParB proteins and consequently have broad significance. The authors show that the spreading activity is dependent on the arginine-rich motif in ParB, previously demonstrated for *B. subtilis* ParB (Graham, 2014), for example. They examine a chromosomal ParB from *V. cholera*, which binds 3 *parS* sites on Chromosome I, and the data also agree with their modeling.

Overall I think that the experiments are convincing and thorough, and this is an important contribution to our understanding of ParB complex assembly *in vivo*.

Major comments:

1. Fig 2 and complex size as a function of ParB concentration. The authors use modeling and ChIP-seq patterns to argue that size of the complex does not vary but the ParB density does change at different ParB concentrations. In the modeling, size is the sigma variable, which was determined experimentally from super-resolution microscopy. Have they tested the prediction by SR microscopy at different ParB concentrations? I think this would be an important experiment to include if feasible. These data would strengthen the conclusion if they agree with the prediction (and would be a problem if they did not).
2. Supplementary Fig S3D and E: The analyses presented are not explained in the main document (and in fact S3E is not mentioned anywhere). Fig S3D is referenced only in the Discussion (pg 18), to support no binding outside of the *parS* region, and I find the modeling confusing (and likely a general reader would also). Why do they simulate different DNA fragment sizes when they appear to know the average fragment size of their library? They state "Here, the modeling describes only the ParB DNA binding on the 10 specific *parS* sites." How does this relate to the math they present in the paper for the caging model? Which parameters are not included? It seems to me that they could explicitly tell us which parameters are excluded or set to zero to obtain the simulations. If they include this figure the rationale and results should be discussed in the main paper.
3. The authors could address or discuss the "silencing" property of spreading, which is of general interest since silencing was the first reported manifestation of extensive ParB binding and is still referenced. Silencing occurs primarily or only when ParB is overexpressed (and is not necessary for partitioning), but since the authors show that ParB binding patterns do not vary with concentration, why does silencing occur? Is it a function of ParB density? It would be interesting to at least speculate in the Discussion, since the authors have done this comprehensive analysis at high ParB concentrations.

Minor comments:

1. Fig 2: Normalization could be described better here in the legend. The explanation in Methods is somewhat technical. I assume that absolute binding is different in each curve because protein

concentration is higher (and density is higher as above), so this binding is relative. What is it normalized to?

2. pg 1, line 8: Delete "cytoskeletal" - this is a vestige of early models proposing that Walker ATPases might work as structural filaments (such as the actin-like ParM ATPase), and since the authors support the current view that they do not (pg 1 "a reaction diffusion-based mechanism"), it is not clear why this term is still used here.

Reviewer #3:

In this manuscript, Diaz et al (2018) described a set of experiments to confirm the "nucleation & caging" model for ParB-parS nucleoprotein formation, specifically for the F-plasmid system. Later, they extended the computational model, together with in vivo ChIP-seq for Vibrio ParB, to suggest that the "nucleation & caging" model is widely conserved for both plasmid and chromosomal ParB-parS systems. In general, I endorse this manuscript because it provides an alternative model to the popular "spreading & bridging" model (Graham et al 2014). However, the current form of this manuscript is not much of a conceptual/experimental advance compared to their previous manuscript that first described the "nucleation & caging" model (Sanchez et al 2015). I found the Vibrio data most interesting, but they are weak and does not contribute much to their model (see below). I listed some of the comments below for the authors to consider.

Major comments:

- 1) The experimental procedures and supplemental procedures are lacking a lot of details. There is no information on experimental replicates either. There is a complete lack of details on the data analysis, computational model, and fitting of data to models. Given that this manuscript relies heavily on fitting experimental data to computational models/simulations, the complete lack of details here made it extremely hard to judge this manuscript.
- 2) Did Diaz et al perform ChIP-seq replicates? There is no information on biological replicates for any experiment in this manuscript at all. There is no mention on negative controls for any ChIP-seq experiments. Given that ChIP-seq were performed using a polyclonal antibody to His-ParB, a negative control is required.
- 3) Fig 2A (and 2A inset) and page 9: why rescale for 0.04 and 0.016? I do not understand what rescale means here, again because the experimental procedure for this part was missing. I do not quite understand the rationale for rescaling here either. Without rescaling, the ChIP-seq line for 0.04 and 0.016 does not fit their favorite model of "nucleation & caging" at all but rather fits the "1D-spreading" model?
- 4) Page 10 and 11 about the claim that "the size of the dynamic ParB/parS cluster is independent of ParB intracellular concentration": I do not see a direct measurement of the size of the ParB/parS cluster/focus here. Diaz et al seems to infer the size from ChIP-seq data. I do not think the 1D ChIP-seq signal can tell about the 3D physical size of the ParB/parS cluster/focus/cage, only microscopy data can.
- 5) Page 11 about the formation of B'2 and B'3 secondary complexes. Are B'2 and B'3 due to the presence of more than 1 parS site in the DNA probe for EMSA? Recently, the C-terminal domain of Bacillus ParB was shown to bind DNA non-specifically and contributes to ParB/parS nucleoprotein complex formation in vitro and in vivo. Is this also the case for F-plasmid ParB C-terminal domain? The pattern of B'2 and B'3 formation is reminiscent of non-specific DNA-binding activity of Bacillus ParB.
- 6) The ChIP-seq experiment for Vibrio ParB lacks negative controls. Also where did the authors get all the parameters for constructing the simulation for Vibrio ParB? This is leading back to my point that details of experimental procedures, especially for the computation model/simulation, is completely missing.
- 7) Page 14-15 about the claim that "nucleoprotein, not active transcription, are major determinants for the impediment of ParB stochastic binding". All the experiments here can only say that active transcription does not impede ParB stochastic binding. There is no direct experiment about nucleoprotein complex. The claim that nucleoprotein complex impedes ParB stochastic binding were inferred secondarily from other experiments. If the authors want to make this claim, they should do a more direct experiment, for example insert a small tetO/lacO array and express TetR/LacI. If the authors do not want to do these experiments, they should remove this claim from

this section and from the Discussion too.

8) Diaz et al claimed that ParBF-3R is not mis-folded/prone to misfolding because it can still bind parS. A binding to parS by a purified protein is not a concrete evidence for WT-level folding. ParB G77S from *Bacillus* binds to parS just as well as WT but has some folding caveats (Song et al 2017). If Diaz et al wants to make this claim, they need to do CD or equivalents to compare their purified 3R mutants to their purified WT protein.

Minor comments:

- 1) A supplementary table detailing their ChIP-seq data (number of reads sequenced, mapped, and repeats) are needed.
- 2) A better description that distinguishes the "nucleation & caging" from the "spreading & bridging" model in the Introduction is needed. From reading their Introduction, I cannot tell the difference. Both models rely on specific ParB-parS binding, on some degrees of ParB binding to non-specific DNA, and on ParB-ParB interactions. What is the difference here?
- 3) Page 2-Abstract-the 3rd sentence from bottom: Caging should be lowercase.
- 4) Page 18: *S. Venezuela* should be *S. venezuelae* (italicized).

1st Revision - authors' response

13th September 2018

Point-by-point response to reviewers' comments.

Color-code: Text quoted from reviewers' comments are in black, our answers are in green and the changes to the manuscript are shown in red (also written in red in the manuscript).

We also have made some other modifications in the text, indicated in blue, that correspond to changes in the format as requested by the journal or to our own corrections.

Reviewer #1:

In this work, the authors provide further evidence to support their Nucleation and Caging model of partition complex formation and argue that the mechanism is widely conserved. The main results are:

-The ParB density profile (as a function of genomic distance from parS) for different intercellular ParB concentrations collapse onto one another. This is not captured by previous models but is by their model if the size of the partition complexes (ParB foci) is independent of the intercellular ParB concentration.

-Residues in the N-terminal domain are required for partition complex formation but not parS binding.

-Nucleoprotein complexes, but not active transcription, impede ParB binding.

ParB complexes are dynamic and turn over on a 1 min timescale.

-Consistent ParB profiles are observed when parS is inserted into the chromosome and in around the native chromosomal parS sites of *V. cholera*.

These results are generally well supported by the data and would make a useful contribution to the field.

However, I have some comments that I believe should be addressed before publication.

We thank the referee for his/her careful reading of the manuscript and his/her positive criticism. We have addressed the issues raised below.

Major points:

- In the authors' model, ParB binds and unbinds DNA and itself stochastically. There is no mention of 1D spreading. Yet they show that nucleoprotein complexes act as a barrier to ParB spreading (eg. incC, locus A and B in Fig. 1). If ParB is dynamic, why does its density not recover after the obstacle and thereby produce an approximately symmetric binding pattern? Indeed, in stationary phase, in which transcription is greatly reduced, the authors nicely show that the signal is much more symmetric.

In our model, there is no 1D-spreading at play. As we proposed in our previous manuscript (Sanchez et al., 2015, Cell systems), the strong "barrier" effect leading to an asymmetry in ParB binding pattern would rather be due to an increase in spatial distance away from *parS*, which strongly

reduces the probability that the DNA after the “obstacle” interacts with the confined ParB cluster nucleated at *parS*. Indeed, on the F plasmid, a large nucleoprotein complex assembles on *incC* and *ori2* by binding of several RepE dimers to arrays of iterons that then interacts with each other (Das and Chattoraj, 2004). We previously proposed, and discussed in the current manuscript p. 17-18, that this large complex would be therefore excluded from the vicinity of *parS* thus reducing the probability that ParB binds to sequence beyond these two arrays.

In the case of the chromosomally-inserted *parS_F*, the nucleoprotein assembled on the locus A (and the others) is much smaller than the RepE-iterons complexes. Also, the ParB signal is higher in Stationary phase, certainly due to the lower overall intracellular dynamic as we discussed in the current discussion section. This increase in the signal to noise ratio allows an easier detection of the ParB signal at long distance from *parS*, highlighting the recovery beyond the locus A. After normalization of the signal (Fig. 5C), we show that this recovery is also present in exponential phase and is still compatible with the characteristic power law distribution. Such a behavior is only described by the ‘Nucleation and caging’ model.

We had a sentence in the legend of Fig 5C: “These data are not in favor of the ‘1D-spreading’ or the ‘Spreading and bridging’ models that predicts a basal uniform distribution or a linear decrease after a barrier, respectively (Broedersz et al., 2014).”

Connected with this, the statement that 'Nucleoprotein complexes, but not active transcription, are the major determinants for the impediment of ParB stochastic binding' is, to me, misleading. Does the data (exp vs exp+rif vs stationary, Fig. 5) not indicate that RNA polymerase acts as roadblock, whether involved in active transcription or stalled by rifampicin?

We totally agree with the reviewer on this remark that stalled RNA polymerases (upon rifampicin treatment) still act as a “barrier”. We have therefore considered that stalled RNA polymerases on promoters are static nucleoprotein complexes, and because no difference in ParB binding signal was observed in the presence of rifampicin, we proposed that the action of transcription *per se* is not acting as barrier, but RNA polymerases loaded or stalled at the promoter impede ParB binding. This is the only conclusion that was intended to be made.

To clarify our statement, we replaced the sentence p. 14 “This indicates that active transcription by RNA polymerase is not a major impediment to ParB binding” as follows:

“This indicates that active transcription by RNA polymerase is not a major impediment to ParB binding, but rather that RNA polymerases bound or stalled at the promoter could”.

Might gene orientation also play a role? The authors could check how gene orientation correlates the position of peaks and dips. It would be interesting to know whether this would indicate some role for 1-D spreading.

This is an interesting point but we think that it goes beyond the scope of the current study on the assembly mechanism of ParB partition complexes, as this would rather explore how ParB clusters sense its local genomic environment. It will require to construct and modify several loci in isogenic strains and to assay them by ChIP-sequencing. From our current data, we observed that dips frequently corresponded to the promoter location regardless of orientation, as mentioned p. 14: “The major dips in the ParB_F DNA binding signal are often found at promoter loci (Fig. S1A).”

No change.

- The authors' model makes an interesting and testable prediction i.e. that the size of ParB foci is independent of the intracellular concentration. This perhaps could be tested experimentally using standard fluorescence microscopy (in the model the authors use a value coming from super-resolution microscopy) and would provide stronger support for their model. I realise that additive effects may mean that a brighter focus appears wider than it actually is but such effects may not be significant over a moderate range of intensities and it may be worth a try.

This prediction could not be tested by standard fluorescence microscopy. Indeed, the maximal size of the ParB cluster (< 75 nm) is much lower than the diffraction limited resolution of microscopy (~300 nm). The increase of ParB level by 10 fold will only increase the cluster size by ~2-fold if the concentration of ParB remains constant in the sphere, thus still below the resolution. Nevertheless, we had thought to this issue by applying superresolution, namely 3D-SIM and PALM. However, in both cases, limitations still apply. After performing 3D-SIM (resolution x,y,z of 90x90x270 nm), we found that the apparent size of clusters maintained a range of 120-180 nm and was independent of the expression (with varied intensities) preventing any conclusion. In PALM, the technique we

previously used to estimate the maximal cluster size, the limitation arises from the time required to acquire all images on live cells (total acquisition time was ~15 minutes; Sanchez et al., 2015). During this period of time, the cluster is moving around and thus, even if highly confined, the size of the cluster appears larger than it is. These technical limitations have thus prevented us to address this question experimentally.

No change.

- The description of the Monte Carlo simulations is severely lacking. All I could find was in the legend to Fig. 1. This should be explained in full and it should be made clear how ParB interacts with the polymer interacts and the effect thereon. Related to that, the model includes ParB-ParB interactions but it wasn't clear to me if ParB is treated explicitly in the simulations or simply enforced externally as a fixed spatial gradient.

We thank the Referee for this remark and apologize for the lack of description. In this work, we are using a simple, phenomenological and semi-quantitative approach to show on general physical ground that the only mechanism able to account for the long range decay observe in ChIP-seq data is the thermal fluctuation of a free polymer in a sphere of high concentration of ParB. More precisely, the polymer is simulated explicitly but the particles are not. The polymer is put into contact with an 'abstract' reservoir of particle with a Gaussian probability distribution centered on *parS* – thus the gradient of ParB is forced externally. During a Monte Carlo step, binding sites of the polymer are checked randomly and are populated with a protein according to the probability written in the new version of the MS.

We have now added more descriptions and details on the fitting procedure and Monte Carlo simulations in the Materials and Methods section. We also provided in the Expanded view (Fig. EV2) full details for the analytical calculations.

Lastly, we changed the main text, p. 7, as follows: “We modeled the DNA molecule by a Freely Jointed Chain (FJC) constituted of N monomers of size a (Kuhn length about twice the persistence length of the corresponding Worm-like chain (Schiessele, 2013)). One particle is always attached on *parS* whereas non-specific sites are in contact with a reservoir of particles displaying a Gaussian distribution centered on *parS*. The ParB density was normalized to 1 by the value on the right side of *parS*, and captured for non-specific sites in the following phenomenological formula as the product of two probabilities integrated over the volume:”

Minor points:

- How much IPTG is used in Fig 1E?

We used 100 μ M IPTG in this experiment. We added this precision in the legend of Fig. 1E as follows: “Cells were grown in the presence of 100 μ M IPTG.”

This information is also added in the Expanded view Table. EV1 that summarizes all the ChIP-sequencing dataset.

- As the inset in Fig. 2A is arguable more important than the outer figure, the authors may want to swap the two around.

We thank the reviewer for his/her suggestion and agree that the main information is the invariance of the ParB signal with a rescaling value, which was previously the information in the inset. We have now swapped the inset and main graphs, and changed accordingly the figure legend.

Perhaps correlation analysis would help demonstrate the locations of peaks and dip coincide.

We agree that a correlation analysis would provide quantitative information on the observation that “the dips and peaks in the pattern are highly conserved”.

We have now made such analyses on the right sides of *xylE::parS_F* in the various conditions tested. We have also compared these patterns with the one from the plasmid F and from a theoretical power law curve. We observed lower levels of correlation in these two latter conditions (0.81) than from all other conditions (> 0.93) confirming the observations that dips and peaks are similarly located and conserved in a given genomic environment.

We now present and explain these new analyses in the Expanded view Table EV2, and we added the following sentence in the main text p. 9: “, as confirmed by correlation analyses (Table EV2).”

- Is the orange line in Fig. 2A the same as in Fig. 1E?

Yes, the orange line is the condition at 100 μM IPTG and is the same data as in Fig. 1E. We have now explicitly indicated it in the legend of Fig. 2A as follows: “Note that the ChIP-seq data at 100 μM IPTG induction (16) are the same as in Fig. 1E.”

- It is stated in the text that the densities plotted in Fig. 2A are only of the right side of profile. This should be stated in the legend and it seems to also be the case for Fig 1C,E and Fig S1 but is not stated.

This information was only provided in the legends for Fig. 1C & E: “is displayed over 14-Kbp on the right side of $parS_F$ ” and in Fig. 2A “ChIP-seq density on the right side of $parS_F$ ”. We have now added this precision in Fig. S1 as follows: “is displayed on the right side of $parS_F$ over 14-Kbp as in Fig. 1E”

- Why do the 3 curves with lowest ParB ratio not have the dip at 9kbp in Fig. 2A?

The dip at 9kbp is not visible for these three conditions because the signal for these patterns reached the basal level around this genomic position. This reduction in specific signal is expected for the lower levels of $ParB_F$ available for assembling the clusters. By rescaling the curves for direct comparison, it results in an increase of the basal level for these three curves. The major point is that the shape of the curves is strikingly superimposable and still compatible to the characteristic power law. This behavior makes a clear discrimination with the different predictions of the three models. We added a note in the legend to explain it as follows: “Note that the dips at $\sim 9\text{-kbp}$ are not visible for the low levels of available ParB since the signal is close to the basal level.”

- p14, p16 The authors should be careful in their wording regarding Power law fitting. It is one thing to say the analytic (power law) relation is consistent with the data and it is another to say that the data exhibits a power law behaviour. It is very easy to obtain a good fit of experimental data to a power law and caution is required. See Stumpf & Porter, Science 2012, DOI: 10.1126/science.1216142

We totally agree with the Referee. We are now more careful in our wording and have changed the text accordingly pp.7, 14 and 16, by writing that “The characteristic asymptotic decay is compatible with a power-law”

- Reference for the form of $P(r,s)$ in equation 1.

The form of $P(r,s)$ contained actually a typo in the prefactor. This has now been corrected in the new version and we provided an expanded view figure (Fig. EV2) to explain the calculation. We added the following sentence: “(see Fig. EV2 for the details of the calculation)” and a reference (de Gennes, 1979) in the main text.

- 'F plasmid' instead of 'plasmid F'

Both wording are found in the literature. Nevertheless, we agree to consistently use only the most frequent one throughout the manuscript. We changed all occurrences to **F plasmid**.

- p10. 'model predicts a probability ..' -> 'model predicts a density ..'

Both terms can be equally used because the local density per site equates the probability to find a particle on this site: both take the same value comprised between 0 and 1.

We changed to “density per site”

- p13 What is the Stochastic Binding model? Is it the same as the Nucleation and Caging model? One name would suffice.

We agree that using two names is confusing. We now refers only to ‘Nucleation and caging’ for the model.

We also changed our nomenclature for the probability of binding P_{SB} to P_{NC}

- 'stochastic binding'. All binding is stochastic, whether slow or fast. This is not a useful phrase. I think the authors rather mean that binding/unbinding occurs on relatively fast timescale i.e. foci are not static but are turned over continuously.

ParB stochastic binding refers to a simple binding and unbinding process on nsDNA by contrast to the cooperative ParB-ParB (dimer-dimer) energetic mechanism proposed in '1D-spreading' and 'Spreading and bridging' that both requires 1D-polymerisation of ParB onto the DNA. Stochasticity also refers to the behavior of the DNA molecule, modeled by a freely fluctuating polymer under thermal fluctuations, which allows any sequence at a given genomic distance from *parS* to return back into the ParB cluster nucleated at *parS*. We have defined this concept in the introduction as follows: "This model therefore proposes that the DNA surrounding the *parS* site interacts stochastically with the sphere of high ParB concentration", and in Fig. 1A: "The DNA entering the cluster is bound stochastically by ParB".

No change.

- p15. That 90% of ParB are in clusters, does not at all imply that partition complexes are stable structures. They could still be turned over slowly or rapidly.

We agree that our oversimplified argument reads incorrectly with the reference to an undefined "stable structure". We simplified the rational and removed the terms "stable structure". We changed the text as follows:

"However, stochastic binding of most ParB_F on non-specific DNA suggests that partition complexes are highly dynamic. To unravel ParB_F dynamics, we performed ..."

Reviewer #2:

This study examines the in vivo mode of assembly of plasmid and chromosomal partition complexes in bacteria. It has been known for a long time that many ParB protein molecules bind at and around the *parS* partition site, form large complexes that are visible as foci by different fluorescence imaging approaches, and spread many kb outward into non-specific DNA away from the *parS* sites by ChIP and ChIP-seq approaches. The authors have previously proposed the "nucleation and caging" model, and here they further address this model in relation to a second "spreading and bridging" model (the 1D spreading models originally proposed have been previously ruled out by these and other authors). The authors have refined the caging model, and a new contribution in this work is that they test it by examining ParB patterns as a function of intracellular ParB concentration (F plasmid ParB), both experimentally and mathematically. They find that the patterns of ParB binding/spreading are independent of ParB concentration, which fits their modeling of caging "robustly" and best. Another new contribution is an examination of the "dips" in ParB binding on the spreading ChIP-seq curves. It is known that strong protein binding sites provide "roadblocks" to ParB spreading, but the curves outside of the roadblocks are not smooth. Some dips correlate with promoters, and the authors show that (i) they do not need active transcription, and (ii) are absent when the promoter region is removed, for example.

The study also confirms ParB spreading properties that have been examined in other systems, which, while not new, are important contributions to show that these properties are general ones for these types of ParB proteins and consequently have broad significance. The authors show that the spreading activity is dependent on the arginine-rich motif in ParB, previously demonstrated for *B. subtilis* ParB (Graham, 2014), for example. They examine a chromosomal ParB from *V. cholera*, which binds 3 *parS* sites on Chromosome I, and the data also agree with their modeling.

Overall I think that the experiments are convincing and thorough, and this is an important contribution to our understanding of ParB complex assembly in vivo.

We thank the referee for his/her careful reading of the manuscript. We have addressed his/her comments and suggestions that we believe improve the clarity of our work and reinforce our findings.

Major comments:

1. Fig 2 and complex size as a function of ParB concentration. The authors use modeling and ChIP-seq patterns to argue that size of the complex does not vary but the ParB density does change at different ParB concentrations. In the modeling, size is the sigma variable, which was determined experimentally from super-resolution microscopy. Have they tested the prediction by SR microscopy at different ParB concentrations? I think this would be an important experiment to include if feasible. These data would strengthen the conclusion if they agree with the prediction (and would be a problem if they did not).

This comment has been fully addressed in response to the second main point of reviewer #1. As indicated, we have tried 3D-SIM but the size of the cluster is below the limit of resolution preventing any conclusion.

2. Supplementary Fig S3D and E: The analyses presented are not explained in the main document (and in fact S3E is not mentioned anywhere). Fig S3D is referenced only in the Discussion (pg 18), to support no binding outside of the *parS* region, and I find the modeling confusing (and likely a general reader would also). Why do they simulate different DNA fragment sizes when they appear to know the average fragment size of their library? They state "Here, the modeling describes only the ParB DNA binding on the 10 specific *parS* sites." How does this relate to the math they present in the paper for the caging model? Which parameters are not included? It seems to me that they could explicitly tell us which parameters are excluded or set to zero to obtain the simulations. If they include this figure the rationale and results should be discussed in the main paper.

We agree that this information is confusing and that we did not provide explanation to allow understanding of the rationale that underlies this specific modeling. These figures and specific modeling were intended to determine whether the residual signal outside *parS_F* observed with the ParB_F3R* mutant arise from residual dimer-dimer interaction or if it could be explained solely by the size of the DNA fragments to which ParB is bound. It does not relate to the math of the 'Nucleation and caging' model but to the binding of a protein to its specific site for which the output is directly link to the size of the DNA fragment in the library. For the simulation, we have tested several DNA fragment sizes surrounding the measured one to display how this parameter changes the resulting profiles, and thus to be able to conclude that no ParB binding to non-specific DNA binding was detected with the mutant.

We have now included the following statement in the main text p. 12 in relation to these figures (now fig. S3E and F): "Indeed, no residual ParB_F binding to non-specific DNA was detected when the size of the DNA fragments in the IP library is taken into account (Fig. S3E)" and "The same patterns was also observed with ParB_F-3R*-mVenus (appendix Fig. S3C and S3F)".

We also describe this specific modeling in the Materials and Methods section, and accordingly simplified the figure S3E legend as follows: "Here, the modeling describes only the ParB specific DNA binding on *parS* sites (see Materials and Methods)".

3. The authors could address or discuss the "silencing" property of spreading, which is of general interest since silencing was the first reported manifestation of extensive ParB binding and is still referenced. Silencing occurs primarily or only when ParB is overexpressed (and is not necessary for partitioning), but since the authors show that ParB binding patterns do not vary with concentration, why does silencing occur? Is it a function of ParB density? It would be interesting to at least speculate in the Discussion, since the authors have done this comprehensive analysis at high ParB concentrations.

We agree with the reviewer that this is an interesting point to discuss. As mentioned by the reviewer, we think that higher density of ParB would reduce the access of the RNA polymerase to the ParB cluster. Notably, a reduced accessibility of DNA gyrase to the vicinity of *parS* has been already proposed (Bouet et al., 2009, JBC) to explain the ParB/*parS*-dependent reduction of negative supercoiling observed on mini-F plasmids (Biek and Shi, 1994).

We now discussed this point as follows p. 17: "In the case of F and P1 plasmids, overexpression of ParB was reported to silence genes in the vicinity of their cognate *parS* (Lynch and Wang, 1995; Lobočka and Yarmolinsky, 1996), by a mechanism based on 1D-spreading (Rodionov et al., 1999). Our finding that the size of ParB clusters is invariant but their density increases with ParB overexpression provides a new explanation for the silencing phenomenon. We propose that RNA polymerases accessibility to promoters present near *parS* is dependent on the ParB density within the cluster. At a physiological ParB level, RNA polymerase would have efficient access to promoter

sites while upon the rise of ParB level their diffusion within the high density cluster would be reduced proportionally to the overexpression level, as observed experimentally (Rodionov et al., 1999). This is reminiscent of the observation that a change in the level of supercoiling is specifically induced on ParB-*parS* carrying mini-F plasmids (Biek and Shi, 1994). It has been shown that this deficit in negative supercoiling could be due to the reduced accessibility of DNA gyrases to the small sized mini-F plasmid (< 10-kbp) that is “masked” by the ParB-*parS* nucleoprotein complex (Bouet et al., 2009). The invariance in the size of the ParB cluster but the density may also well explain the supercoiling deficit observed *in vivo*.

Minor comments:

1. Fig 2: Normalization could be described better here in the legend. The explanation in Methods is somewhat technical. I assume that absolute binding is different in each curve because protein concentration is higher (and density is higher as above), so this binding is relative. What is it normalized to?

We apologize for the lack of precision in Fig. 2 regarding to the normalization. Normalization was performed as in Fig. 1C and 1E. We have now rephrased the legend as follows: “..., normalized as in Fig. 1C, E with the amplitudes of the curves rescaled by the indicated factors (1.2, 10 or 50) to overlap with the curves of highest amplitude”.

We have also added some detail in the legend of Fig. 1C to be more precise. The sentence now reads as follows: “The ParB density, normalized to 1 at the first bp downstream the last *parS_F* binding repeat after background subtraction, is displayed over 14-Kbp on the right side of *parS_F*.”

We also modified the section “ChIP-sequencing assay and analysis” of the Materials and Methods, as follows: “Cognate Input and IP samples were normalized by the number of total reads for direct comparison. For the ParB density plots, the data were normalized after background subtraction and set to the value of 1 at the last bp of the 10th repeat of *parS_F*, allowing to display the results of Monte Carlo simulation on the same graph.”

2. pg 1, line 8: Delete "cytoskeletal" - this is a vestige of early models proposing that Walker ATPases might work as structural filaments (such as the actin-like ParM ATPase), and since the authors support the current view that they do not (pg 1 "a reaction diffusion-based mechanism"), it is not clear why this term is still used here.

We agree with this remark and have removed “cytoskeletal”.

Reviewer #3:

In this manuscript, Diaz et al (2018) described a set of experiments to confirm the "nucleation & caging" model for ParB-*parS* nucleoprotein formation, specifically for the F-plasmid system. Later, they extended the computational model, together with *in vivo* ChIP-seq for Vibrio ParB, to suggest that the "nucleation & caging" model is widely conserved for both plasmid and chromosomal ParB-*parS* systems. In general, I endorse this manuscript because it provides an alternative model to the popular "spreading & bridging" model (Graham et al 2014). However, the current form of this manuscript is not much of a conceptual/experimental advance compared to their previous manuscript that first described the "nucleation & caging" model (Sanchez et al 2015). I found the Vibrio data most interesting, but they are weak and does not contribute much to their model (see below). I listed some of the comments below for the authors to consider.

We thank the referee for his/her careful reading of the manuscript. We have carefully taken into account her/his comments and suggestions that we believe improve the clarity of our work and reinforce our findings.

Major comments:

1) The experimental procedures and supplemental procedures are lacking a lot of details. There is no information on experimental replicates either. There is a complete lack of details on the data analysis, computational model, and fitting of data to models. Given that this manuscript relies heavily on fitting experimental data to computational models/simulations, the complete lack of details here made it extremely hard to judge this manuscript.

We apologize for the lack of details in the procedures we performed. We have now added (i) an Expanded View (Fig. EV2) for the details of the analytical calculation, (ii) three sections in the Materials and Methods for the Fit of the parameters and Monte Carlo simulations, for the modeling of the ChIP-seq data with the integration of the average fragments size of the DNA library, and for the new correlation analyses we performed, and (iii) a technical description of the Monte Carlo procedure in the Appendix Materials and Methods.

We also provided a synoptic table (Table EV2) for the details on ChIP-sequencing experiments.

2) Did Diaz et al perform ChIP-seq replicates? There is no information on biological replicates for any experiment in this manuscript at all. There is no mention on negative controls for any ChIP-seq experiments. Given that ChIP-seq were performed using a polyclonal antibody to His-ParB, a negative control is required.

We thank the reviewer for these remarks and apologize for the lack of information.

We did not performed ChIP-seq replicates for all experiments due to the cost of this technique. However, most of the ChIP-seq assays have been performed in several conditions or strains or with a variant allele, which we think could account for replicates (even if not performed in parallel the same day with a second clone and in the same exact condition) as detailed below in the order of appearance in the manuscript:

- in the case of the 100-kbp plasmid F, we observed similar results (Fig. 1C) as the one we previously published using a smaller version of the F plasmid (Sanchez et al., 2015). This is already mentioned in the main text.

- In the case of parS_F inserted on the E. coli chromosome, we performed Chip-seq in a strain where ParB_F is produced from a chromosomal locus, with the two orientations of parS_F ($\text{xylE}::\text{parS}_F$ in DLT2075; Fig. 1E) and ($\text{xylE}::\text{parS}_F\text{rev}$ in DLT3491; Fig. S1B-C) with similar results. To function as another “replicate” from $\text{xylE}::\text{parS}_F$, we have now added in the manuscript two ChIP-seq assays performed with ParB_F or $\text{ParB}_F\text{-mVenus}$ provided *in trans* from plasmids, strains DLT3567 and DLT3548, respectively, which also displayed the same results (Appendix Fig. S1D and S3D).

We now included the following sentence p. 6: “Also, similar patterns were observed when ParB_F or $\text{ParB}_F\text{-mVenus}$ were expressed *in trans* from a plasmid (appendix Fig. S1D and Fig. S3D)”. We also mentioned this new experiment in the legend of Fig. 1E as follows: “Note that a highly similar ParB_F DNA binding pattern is obtained when ParB_F was expressed *in trans* from a plasmid (strain DLT3567; Appendix Fig. S1D)”.

- In the case of the variation of the level of ParB_F available to assemble in clusters (Fig. 2A), we do not have performed duplicates since we have preferred assaying two overexpression (16, 28) and two titration (0.04, 0.016) conditions which are coherent altogether.

- In the case of the $\text{ParB}_F\text{-3R}^*$ variant, we presented two ChIP-seq assays that give the exact same result (binding only to parS_F and not to the neighboring sequences), one with the $\text{ParB}_F\text{-3R}^*$ (DLT3726; Fig. 3C) and the other with $\text{ParB}_F\text{-3R}^*\text{-mVenus}$ (DLT3566; Fig. S3B). They are both expressed *in trans* from a plasmid similar to the one used in Fig. S1D to produce ParB_F WT. We added in the legend of Fig. 3C the following sentence: “Note that a highly similar DNA binding pattern is obtained with $\text{ParB}_F\text{-3R}^*\text{-mVenus}$ (strain DLT3566; Appendix Fig. S3C).”

- In the case of the *Vibrio cholerae* ChIP-sequencing, we have performed two independent replicates (cultures on different days and separate ChIP-assays). This is now reported on the synoptic Table EV2 as replicates R1 and R2. They were both included in the Geo database under the number GSE114980 mentioned in the main text in the section “Accession number”.

- In the case of the Rifampicin treated DLT2075 cells, we have performed two independent replicates. It is now listed in Table EV2 and indicated in the legend of Fig. 5, as follows: “The assays have been performed in duplicate for the +Rif...”.

- In the case of the ChIP-seq performed on the $\text{xylE}::\text{parS}_F$ strain in stationary phase, we do not have a duplicate experiment in this condition. We have now indicated it in the legend of the Fig 5: “The assays have been performed {...} once for the stationary phase experiments.”

- In the case of the D (locus A) strain, after checking the file and data (thanks to the reviewer) we noticed a mistake: one of the replicate (R1) was not performed with the correct strain. We have now removed it from our list and we are in the process of removing the corresponding files (Input + IP) from the Geo database # GSE115274. We now indicated in the legend of Fig. 5D: “The assay in the D (locus A) genomic context has been performed once.”

We apologize also for the lack of information on control experiments. First, we now provided the missing data for the input samples which are indeed a major information that we omit to display in the final figures. We added these data in Fig. 1 C, Fig. 1E and Fig. 4B. It clearly reveals the high level of specificity in our ChIP-sequencing assays. Also, ParB_F binding signal is only observed at and around *parS_F* and not elsewhere on the genome showing the strong ParB-*parS* specificity. We forgot to recall this explicitly in the main text. We now added the following sentence p. 6 : “Besides the strong ParB binding enrichment in the vicinity of *parS_F*, no other difference in the pattern between the input and IP samples were observed on the F plasmid and on the *E. coli* chromosome.”. This is also the case for *Vibrio cholerae* data, and we now mentioned it as follows p. 12 : “No other ParB binding was observed over the *Vibrio* genome.”

ChIP-seq assays were performed using a polyclonal antibody raised and affinity-purified against WT ParB_F or a polyclonal antibody raised and affinity-purified against his-tagged ParB_{1Vc}. As mentioned above, we observed in both cases a highly specific ParB binding signal only at *parS* sites, clearly indicating that there is no cross-contamination with other proteins. We added this precision in the Material and Methods section “ChIP-sequencing assay and analysis” as follows: “... , using polyclonal antibodies raised and affinity-purified against WT ParB_F or his-tagged ParB_{1Vc}” In addition, we would like to point out that our experiment with the ParB variant ParB-3R* also provides another level of control on the non-specific DNA binding of ParB in the vicinity of *parS_F*. Indeed, only the *parS* site is occupied by ParB with this allele (strains DLT3726; Fig. 3B).

3) Fig 2A (and 2A inset) and page 9: why rescale for 0.04 and 0.016? I do not understand what rescale means here, again because the experimental procedure for this part was missing. I do not quite understand the rationale for rescaling here either. Without rescaling, the ChIP-seq line for 0.04 and 0.016 does not fit their favorite model of "nucleation & caging" at all but rather fits the "1D-spreading" model?

The reason for rescaling the signal – previously displayed in the Inset and now switched to the main part of Fig. 2A as proposed by Reviewer#1) is to show that the ParB binding signal follows the same distribution pattern at long genomic distance whatever the level of ParB. Indeed, at the very low levels of ParB tested the signal remains high at *parS* (specific site) but faint on non-specific DNA as expected. This representation highlights the invariance predicted by the ‘Nucleation and caging’ model but not by the two others, as presented in the new expanded view Fig. EV2B (previously Fig. S2A).

We add the following precision in the legend of Fig. 2C: “resulting in overlapping prediction profiles with a rescaling of the amplitude corresponding to the WT expression level.”

4) Page 10 and 11 about the claim that "the size of the dynamic ParB/*parS* cluster is independent of ParB intracellular concentration": I do not see a direct measurement of the size of the ParB/*parS* cluster/focus here. Diaz et al seems to infer the size from ChIP-seq data. I do not think the 1D ChIP-seq signal can tell about the 3D physical size of the ParB/*parS* cluster/focus/cage, only microscopy data can.

We did not intend to infer the size of the ParB cluster from ChIP-seq and we agree that such a measurement could only be estimated by microscopy. Here, from our modeling data of the ChIP-seq signal we proposed that the size of the cluster is invariant and that the concentration of ParB within the cluster is varying in relation to the intracellular level of ParB. See our reply to reviewer#1 regarding the current limitation to test experimentally the size of the cluster.
No change.

5) Page 11 about the formation of B'2 and B'3 secondary complexes. Are B'2 and B'3 due to the presence of more than 1 *parS* site in the DNA probe for EMSA? Recently, the C-terminal domain of Bacillus ParB was shown to bind DNA non-specifically and contributes to ParB/*parS* nucleoprotein complex formation in vitro and in vivo. Is this also the case for F-plasmid ParB C-terminal domain? The pattern of B'2' and B3 formation is reminiscent of non-specific DNA-binding activity of Bacillus ParB.

We agree that explanation and a reference on the B'2 and B'3 secondary complexes was lacking. It refers to our previous report, which biochemically characterized the interactions required to form the ParB cluster. Only one *parS* site is present on the DNA fragment (as indicated in the legend of Fig.

3A) leading to the formation of only one specific complex (B1). The two other complexes are forming non-specifically on the DNA fragment and are labelled B'2 and B'3, as defined previously (Sanchez et al., 2015 Cell systems).

For clarity, (i) we modified the sentence p. 12 as follows: "However, in contrast to WT ParB, the formation of secondary complexes (B'2 and B'3), **resulting from non-specific DNA binding and dimer-dimer interaction** (Sanchez et al., 2015), was ...", and (ii) we added the reference (Sanchez et al., 2015) in the legend of fig 3A.

We are aware of the work of M. Dillingham team on their finding that the C-terminal domain of Bacillus ParB was shown to bind DNA non-specifically, and we mentioned it in the introduction p. 4: "The 'Nucleation & caging' model rather proposes that the combination of dynamic but synergistic interactions, ParB-ParB and ParB-nsDNA (Fisher et al., 2017; Sanchez et al., 2015),...". In the case of ParB_F, such activity in the C-terminal domain was not detected as the non-specific DNA binding activity was shown to reside only in the HTH domain (Ah-Seng et al., 2009, JBC). In these two cases, non-specific DNA binding activity was found important to promote the formation of secondary ParB-DNA complexes.

No change

6) The ChIP-seq experiment for Vibrio ParB lacks negative controls.

For *V. cholera*, we had performed a set of preliminary assays prior to ChIP sequencing which was included in a Chapter of a Methods in Molecular Biology book, The bacterial nucleoid (Diaz et al., 2017, MiMB). It displayed optimization steps and control of immunoprecipitation using the ParB1 ChIP-seq of *V. cholera* as an example. We referred to this detailed protocol in the Materials and Methods section. To be more explicit we have now added the following sentences in the new Materials and Methods: "**The optimization step for determining the amount of antibodies needed to pull down all ParBVc-1 in the IP samples was fully described in (Diaz et al., 2017).**"

The anti-ParB_{Vc1} serum was provided by D. Chattoraj that used it previously for ChIP-Chip assay (Baek et al., 2014). In their report, they have controlled for the complete absence of signal in a ParB1 mutant. We do not have reproduced this negative control. Indeed, the only ParB DNA binding signal that we observed is found exclusively on the 3 *parS* sites and surrounding DNA. We have however further purified the serum against purified ParB1_{Vc}-his6 to ensure very high specificity, as now mentioned in the Material and methods.

Also where did the authors get all the parameters for constructing the simulation for Vibrio ParB? This is leading back to my point that details of experimental procedures, especially for the computation model/simulation, is completely missing.

We apologize for the lack of details for these parameters. We now provided a section "**Fit of the parameters**" in the main Materials and Methods section, along with the new Fig. EV2 that describes the analytics calculation.

We also modified the legend of Fig. 4C as follows: "The best fit was achieved with $s=25\text{nm}$ and an amplitude $k=0.15$ leading to $Nt\sim 50$ ParB on the chromosome (see Fig. EV2 for details)."

7) Page 14-15 about the claim that "nucleoprotein, not active transcription, are major determinants for the impediment of ParB stochastic binding". All the experiments here can only say that active transcription does not impede ParB stochastic binding. There is no direct experiment about nucleoprotein complex. The claim that nucleoprotein complex impedes ParB stochastic binding were inferred secondarily from other experiments. If the authors want to make this claim, they should do a more direct experiment, for example insert a small tetO/lacO array and express TetR/LacI. If the authors do not want to do these experiments, they should remove this claim from this section and from the Discussion too.

In the last paragraph of this section we have provided ChIP-seq analysis (Fig. 5D) with a strain in which a region carrying two promoters, two RcsAB and one IHF regulatory binding sites (locus A; Fig. S2A) was deleted. This results in the specific disappearance of the dip at this genomic location. Our conclusion is that the various nucleoprotein complexes, including paused RNA polymerases, that form in this region impedes ParB_F binding.

No change

8) Diaz et al claimed that ParBF-3R is not mis-folded/prone to misfolding because it can still bind parS. A binding to parS by a purified protein is not a concrete evidence for WT-level folding. ParB G77S from Bacillus binds to parS just as well as WT but has some folding caveats (Song et al 2017). If Diaz et al wants to make this claim, they need to do CD or equivalents to compare their purified 3R mutants to their purified WT protein.

We agree with the reviewer that we could not exclude that the protein is partly misfolded. We have not performed CD analyses to address specifically this point. Here, our only point was to show that this ParB_F mutant was fully expressed and that it retains the *parS* specific DNA binding activity, which it does *in vivo* at a level undistinguishable from WT as determined by ChIP-sequencing.

To avoid this overstatement we have corrected the text as follows:

p. 11, we have removed “no defect in protein folding”. The sentence now read: “..., indicating no defect in *parS*_F binding nor dimerization, a property required for *parS* binding”

Minor comments:

1) A supplementary table detailing their ChIP-seq data (number of reads sequenced, mapped, and repeats) are needed.

We apologize for the lack of information on all our ChIP-sequencing experiments. We have now summarized the important data in a synoptic table displayed in the Expanded view section as **Table EV1** and we referred to it in the main text.

2) A better description that distinguishes the "nucleation & caging" from the "spreading & bridging" model in the Introduction is needed. From reading their Introduction, I cannot tell the difference. Both models rely on specific ParB-parS binding, on some degrees of ParB binding to non-specific DNA, and on ParB-ParB interactions. What is the difference here?

We have added a new expanded view figure EV1 to present the three models in part A and the modeling of the DNA binding profiles in the vicinity of *parS* as a function of the ParB level in part B. It replaces the panel A of the previous Fig. S2A.

We now refer to **Fig. EV1A** in the introduction to better explain the models and to **Fig. EV1B** to describe the different physical predictions of these models.

3) Page 2-Abstract-the 3rd sentence from bottom: Caging should be lowercase.

Corrected

4) Page 18: S. Venezuela should be S. venezuelae (italicized).

Corrected

2nd Editorial Decision

24th September 2018

Thank you for sending us your revised manuscript. We have now heard back from reviewer #3 who was asked to evaluate your manuscript. As you will see below, the reviewer mentions that their concerns have been addressed and think that the study is now suitable for publication.

Before we formally accept the manuscript for publication, we would ask you to address a few remaining editorial issues listed below.

Reviewer #3:

I am happy with this revised version of the manuscript. Thank you.

Corresponding Author Name: Bouet Jean-Yves
Manuscript Number: Molecular Systems Biology
Manuscript Number: MSB-18-8516